# Joint Shadow Generation and Relighting via Light-Geometry Interaction Maps

**Shan Wang** [1,2]     **Peixia Li**[1]     **Chenchen Xu**[1]     **Ziang Cheng**[1]     **Jiayu Yang**[1]
    **Hongdong Li**[1,2]     **Pulak Purkait**[1]
[1]Amazon     [2]Australian National University

## Abstract

We propose Light–Geometry Interaction (LGI) maps, a novel representation that encodes light-aware occlusion from monocular depth. Unlike ray tracing, which requires full 3D reconstruction, LGI captures essential light–shadow interactions reliably and accurately, computed from off-the-shelf 2.5D depth map predictions. LGI explicitly ties illumination direction to geometry, providing a physics-inspired prior that constrains generative models. Without such prior, these models often produce floating shadows, inconsistent illumination, and implausible shadow geometry. Building on this representation, we propose a unified pipeline for joint shadow generation and relighting-unlike prior methods that treat them as disjoint tasks-capturing the intrinsic coupling of illumination and shadowing essential for modeling indirect effects. By embedding LGI into a bridge-matching generative backbone, we reduce ambiguity and enforce physically consistent light–shadow reasoning. To enable effective training, we curated the first large-scale benchmark dataset for joint shadow and relighting, covering reflections, transparency, and complex interreflections. Experiments show significant gains in realism and consistency across synthetic and real images. LGI thus bridges geometry-inspired rendering with generative modeling, enabling efficient, physically consistent shadow generation and relighting.

## 1 Introduction

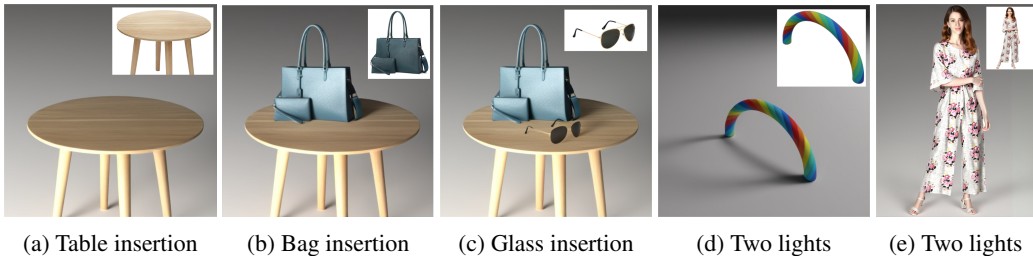

| (a) Table insertion | (b) Bag insertion | (c) Glass insertion | (d) Two lights | (e) Two lights |

Figure 1: Effectiveness of our joint shadow generation and relighting pipeline. Our method produces realistic, texture-aware shadows consistent with object and scene geometry, while preserving faithful relighting across diverse materials such as wood, leather, metal, and glass. (a-c) Multiple object interactions. (d-e) Generalization to multiple light sources.

Shadow generation and relighting are important tasks in a wide range of visual computing applications, including virtual product placement, post-capture image editing, augmented reality, and digital content creation. Realistic shadow generation and relighting require reasoning about how light interacts with scene geometry. Traditional physically based rendering methods achieve this through explicit 3D reconstruction and ray tracing Pharr et al. (2023); Keller (1997), but they are computationally expensive and impractical in single-view settings. On the other hand, recent generative approaches, such as bridge matching Tasar et al. (2024); Chadebec et al. (2025) or diffusion Liu et al. (2020); Hong et al. (2022); Liu et al. (2024a); Winter et al. (2024), can synthesize shadows and

illumination from RGB inputs, but in the absence of physical constraints they often produce floating shadows, inconsistent illumination, or implausible geometry, particularly under complex lighting.

We introduce *Light–Geometry Interaction* (LGI) maps, a novel 2.5D representation that directly encodes occlusion relationships between light and geometry from monocular depth. LGI maps differ from traditional ray tracing, which requires full 3D geometry, by providing a compact, differentiable approximation of light transport suitable for end-to-end learning. They also improve upon prior depth-based conditioning by explicitly coupling depth with light direction, serving as a physics-inspired prior that constrains generative models while remaining computationally efficient.

Building on LGI maps, we design a unified pipeline for joint shadow generation and object-level relighting in scene-aware settings, where a newly inserted object must cast shadows, produce reflections, and be illuminated consistently with the background light. Prior works typically treat these tasks independently: shadow models Liu et al. (2024a); Zhao et al. (2025); Tasar et al. (2024); Chadebec et al. (2025) generate planar or template-based shadows, while relighting methods Zhang et al. (2025a); Kim et al. (2024) focus solely on object reflectance. However, light and shadows are intrinsically coupled — accurate modeling requires reasoning about direct illumination, secondary reflections, and inter-reflections simultaneously. Our joint formulation enforces this coupling, yielding coherent shadow–light interactions that cannot be achieved when the two tasks are separated.

To support this task, we construct a large-scale synthetic dataset, *ShadRel*, for joint modeling of shadows and relighting. Unlike existing datasets that emphasize either hard shadows Liu et al. (2024a); Tasar et al. (2024) or object-only relighting Helou et al. (2020); Kim et al. (2024), ShadRel dataset includes soft shadows, reflective and transparent materials, and inter-reflections, providing a comprehensive resource for training and evaluation.

Through extensive evaluation, we demonstrate that our framework achieves state-of-the-art (SOTA) performance across a wide range of challenging visual scenes, bridging the gap between geometry-free neural rendering and computationally intensive physically-based approaches. Though designed at the single-object and single-light level, it naturally extends to multi-object editing and multiple light sources (Fig. 1). Our framework thus offers an efficient yet physically inspired alternative suitable for practical shadow-aware image editing and realistic relighting.

Our contributions are summarized as follows:

- Light–Geometry Interaction Map: a novel light-aware occlusion representation that bridges the gap between geometry-inspired rendering and unconstrained generative models.
- Joint shadow–relighting pipeline: a unified framework to couple shadow generation and relighting, enabling physically consistent reasoning about direct lighting, secondary reflections, and inter-reflections.
- ShadRel dataset for coupled light transport: a large-scale dataset designed to capture challenging illumination effects, supporting rigorous training and evaluation.

## 2 RELATED WORK

**Shadow Generation.** Classical rendering techniques such as ray tracing Wald et al. (2001); Purcell et al. (2005) and path tracing Christensen et al. (2018); Lafortune & Willems (1996) accurately simulate light transport but require detailed 3D geometry and are computationally expensive. Neural methods attempt to reconstruct geometry from multi-view inputs Zhao et al. (2024); Lin et al. (2023), making them unsuitable for single-view settings. Alternative approaches bypass full 3D reconstruction. SSN Sheng et al. (2021) estimates ambient occlusion from object masks, and pixel height maps approximate geometry for shadow simulation Sheng et al. (2022; 2023). GAN-based models Zhang et al. (2019); Liu et al. (2020) use adversarial training and spatial attention to synthesize shadows, while diffusion-based Hong et al. (2022); Liu et al. (2024a); Winter et al. (2024) and bridge-based methods Tasar et al. (2024); Chadebec et al. (2025) generate shadow regions with adjustable placement. Yet most of these methods are fundamentally 2D, relying on templates or bounding boxes. Hong et al. (2022) decompose synthesis into shape estimation and filling, while Zhao et al. (2025) use rotated boxes and templates. Such heuristics break down in complex scenes with ambiguous occlusion. Depth-conditioned works Griffiths et al. (2022); Kocsis et al. (2024) leverage predicted depth to approximate geometry but require multi-stage pipelines and shading an-

notations. In contrast, we treat predicted depth as a 2.5D structural cue, embedding it directly into LGI maps to provide a differentiable prior for light–shadow interactions in an end-to-end setting.

**Image Relighting.** Relighting methods based on inverse rendering and reflectance models Wenger et al. (2005); Wang et al. (2020); Zeng et al. (2024a); Zhu et al. (2022), primarily focus on regressing radiance on object surfaces, while largely ignoring cast shadows. More recent deep learning approaches Bhattad et al. (2024); Xing et al. (2024); Kim et al. (2024); Liang et al. (2025); Zeng et al. (2024b); Zhang et al. (2025b); Lin et al. (2023); Liang et al. (2025); Zhu et al. (2022) estimate or disentangle image intrinsics (e.g., albedo, normals, roughness, metallic) to constrain relighting, but these typically require strong supervision. Zhang et al. (2025a) instead enforces linear consistency between appearances under different illuminations and their mixture, but requires training scenes with at least two distinct light sources. Some recent work highlights the role of shadows in achieving consistent relighting. Fortier-Chouinard et al. (2024) shows that coarse shadow modeling improves illumination quality, underscoring the need for light–shadow coherence. Still, most prior methods focus on object illumination alone, overlooking the coupled effects of shadows.

**Toward Joint Approaches.** Although shadow synthesis and relighting are tightly coupled in real scenes, very few methods tackle them together. Most follow a sequential design: for example, Griffiths et al. (2022) predicts shadow shape before relighting, and Fortier-Chouinard et al. (2024) uses coarse shadows to guide relighting. These pipelines remain disjoint and rely on handcrafted intermediate steps. Recent generative frameworks such as bridge matching Tasar et al. (2024); Chadebec et al. (2025) and diffusion-based harmonization Winter et al. (2024) offer flexible image-to-image translation, but without explicit light–geometry modeling they often produce inconsistent illumination and floating shadows. In contrast, our method introduces Light–Geometry Interaction maps as a structured prior and integrates them into a single pipeline for joint shadow generation and relighting, capturing secondary reflections and inter-reflections that independent pipelines cannot.

## 3 BASELINE MODEL: LATENT BRIDGE MATCHING

Recent works Winter et al. (2024); Tasar et al. (2024) demonstrate that diffusion models and bridge matching techniques can effectively generate realistic shadows and enable photorealistic image relighting. We adopt latent bridge matching Chadebec et al. (2025) as our baseline model.

Latent bridge matching learns to transform samples from a source distribution $\pi_0$ to a target distribution $\pi_1$, given paired samples $(x_0, x_1) \sim \pi_0 \times \pi_1$. For efficiency, the method operates in latent space using an encoder-decoder pipeline: $z = \mathcal{E}(x)$ and $x = \mathcal{D}(z)$, following Rombach et al. (2022). The method defines a Brownian bridge Revuz & Yor (2013) between latent codes $z_0$ and $z_1$ sampled from their latent distributions, respectively:

$$z(t) = (1 - t)z_0 + tz_1 + \sigma\sqrt{t(1 - t)}\,\epsilon, \quad \epsilon \sim \mathcal{N}(0, I), t \in [0, 1], \sigma \geq 0. \tag{1}$$

where $t$ denotes the time step and $\sigma$ controls noise level. A neural network is then trained to regress the corresponding SDE drift $v_\theta$ along this bridge by minimizing a mean squared error loss, defined as the expectation $\mathbb{E}[\cdot]$ over paired samples $(z_0, z_1)$ and time steps $t$:

$$\mathcal{L}_z = \mathbb{E}\left[\|v(z(t), t) - v_\theta(z(t), t)\|^2\right], \quad v(z(t), t) = \frac{(z_1 - z(t))}{1 - t}. \tag{2}$$

The framework naturally extends to conditional generation by introducing conditioning variables $c$, in which case the SDE drift becomes $v_\theta(z(t), t, c)$. During training, paired samples are encoded to latents, a timestep $t$ is sampled, and noisy samples $z(t)$ are created based on Eq. 1. The predicted target latent is retrieved as:

$$\hat{z}_1 = (1 - t) \cdot v_\theta(z(t), t, c) + z(t), \tag{3}$$

which is then decoded to image space as $\hat{x}_1 = \mathcal{D}(\hat{z}_1)$. The final loss combines latent matching (Eq. 2) with a pixel-level loss $\mathcal{L}_x(\cdot)$:

$$\mathcal{L} = \mathcal{L}_z(\mathcal{E}(x_0), \mathcal{E}(x_1)) + \lambda \cdot \mathcal{L}_x(\hat{x}_1, x_1), \tag{4}$$

where the baseline model adopts LPIPS (Zhang et al. (2018)) as its pixel-level loss. While effective for general image-to-image translation tasks, this baseline framework has limitations for shadow generation and relighting applications. By solely relying on 2D image information, the method lacks

access to crucial geometric properties such as surface normals, depth, and spatial relationships. This absence of 3D geometric understanding limits its ability to accurately model the complex interactions between objects, lighting conditions, and shadow formation, ultimately affecting the physical realism and consistency of the generated results.

# 4 METHOD

Our framework enables accurate and physically informed shadow generation and relighting by introducing LGI information calculated only from monocular depth images. An overview of our approach is shown in Fig. 2. The system transforms shadow-free images $x_0 \in \pi_0$ into shadowed counterparts $x_1 \in \pi_1$. We initialize from a pretrained diffusion model, and keep the encoder $\mathcal{E}(\cdot)$ and decoder $\mathcal{D}(\cdot)$ frozen. Training then focuses on bridge matching conditioned on LGI maps.

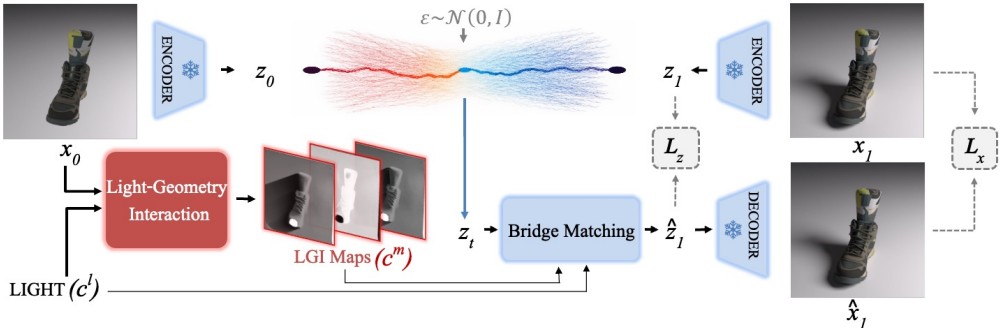

Figure 2: Overview of the proposed method. Our approach uses a bridge-matching strategy to transform shadow-free latent codes ($z_0$) into shadowed counterparts ($z_1$), conditioned on global light cues (e.g., light color, radius) and image-derived light–geometry interaction maps. The key novelty lies in generating these interaction maps from image ($x_0$) and light (see 4.1 for details).

Following Eq. 1, we obtain $z(t)$ and pass it to the network to predict target latent $\hat{z}_1$ as in Eq. 3. The drift network $v_\theta$ is conditioned on $c = \{c^l, c^m\}$, where $c^l$ denotes the global lighting parameters and $c^m$ denotes the LGI maps. The global parameters $c^l$ contain information about light color, radius, distance, intensity, and direction (azimuth and elevation). The azimuth is defined in the camera plane, while elevation is measured relative to it, consistent with our LGI maps construction. The LGI maps $c^m$ provide image-derived lighting-geometry interactions, as detailed in Section 4.1.

To focus computation on the most important regions, we replace the pixel-level loss term $\mathcal{L}_x(\cdot)$ in Eq. 4 with a weighted L1 loss in image space that emphasizes areas of brightness changes:

$$\mathcal{L}_x(\hat{x}_1, x_1) = \frac{1}{M} \sum_{m=1}^{M} w^{(m)} \cdot |x_1^{(m)} - \hat{x}_1^{(m)}|, \quad w^{(m)} = (|x_1^{(m)} - x_0^{(m)}| > \tau) \oplus \mathcal{K}, \quad (5)$$

where $\hat{x}_1$ denotes the predicted shadowed image, $m$ indexes over the $M$ training samples, $\tau$ is a threshold used to identify regions with significant brightness changes, and $\oplus \mathcal{K}$ denotes a dilation operation to expand these regions.

## 4.1 LIGHT-GEOMETRY INTERACTION MAPS GENERATION

In this section, we describe how LGI maps $\boldsymbol{c}^m$ are derived from input images $x_0$ and lighting $\boldsymbol{l}$. Light sources are modeled as point lights, and LGI maps are generated through five steps: depth estimation, 3D lifting, ray sampling, elevation difference calculation, and final map construction.

**Depth Estimation.** We first estimate depth $\boldsymbol{D}$ using an off-the-shelf monocular method Chadebec et al. (2025). Since monocular depth estimation typically produces only normalized depth up to an unknown scale, we rescale the predictions to match the light coordinates. This does not require metric scale—only consistency between light and scene geometry. In ShadRel dataset, light is given in camera coordinates with meter units, while we also provide an image-harmonized variant (see supplementary material) with normalized depth $(0, 1)$ and spatial extent $[-0.5, 0.5]$.

**Lifting 2D to 3D.** Each pixel in the 2D image is lifted into 3D space by using the predicted depth map. Specifically, given the pixel coordinates and the estimated depth, we compute the 3D position in the camera coordinate frame via the inverse camera projection:

$$\boldsymbol{p} = \boldsymbol{D}(u,v) \cdot \boldsymbol{K}^{-1} \begin{bmatrix} u & v & 1 \end{bmatrix}^\top, \tag{6}$$

where $\boldsymbol{D}(u,v)$ is the depth value at pixel $(u,v)$, $\boldsymbol{K}$ is the camera intrinsic matrix and $\boldsymbol{p} \in \mathbb{R}^3$ is the resulting 3D point in camera coordinates. To provide intuition, we use a simple example—a blue ball—in Fig. 3. The diagram shows a cross-sectional view of the 3D space. Because depth maps capture only 2.5D geometry, they fail to provide depth for occluded surfaces. Such ambiguous or occluded regions are marked in pink in Fig. 3.

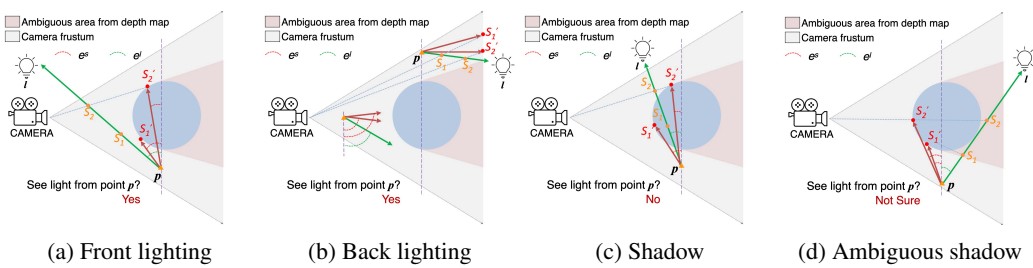

| (a) Front lighting | (b) Back lighting | (c) Shadow | (d) Ambiguous shadow |

Figure 3: Elevation difference calculation. Scene objects are shown in blue. Each 2D pixel is lifted to 3D at point $\boldsymbol{p}$, from which a ray is cast toward the light source $\boldsymbol{l}$. Along this ray, we uniformly sample $n$ points $\boldsymbol{S}$ within the valid front-facing camera frustum. Each sampled point is reprojected onto the image plane to retrieve its depth from the predicted depth map, yielding a set of reprojected points $\boldsymbol{S}'$. The elevation angles $\boldsymbol{e}^s$ of these reprojected points are then compared with the elevation angle $\boldsymbol{e}^l$ of the light ray to compute elevation difference $\boldsymbol{e}^d$. If the light is occluded when viewed from point $\boldsymbol{p}$, that point is likely to lie in shadow.

**Ray Sampling.** As shown in the Fig. 3, a ray is cast from each lifted point $\boldsymbol{p}$ toward the light source $\boldsymbol{l}$. We uniformly sample $N$ points along this ray, forming the set $S = \{\boldsymbol{p} + \delta_n(\boldsymbol{l} - \boldsymbol{p})\}_{n=1}^N$, where $\delta_n$ are evenly spaced scalar distances constrained within the front-facing camera frustum. Each sampled point in $\boldsymbol{S}$ is projected back to the image plane $(u'_n, v'_n)$ to retrieve its corresponding depth from the predicted depth map, and following Eq. 6, resulting in reprojected points $S' = \{S'_n\}_{n=1}^N$. Points with infinite depth are marked as invalid and excluded from further computation.

**Elevation Difference Calculation.** For each valid point in $\boldsymbol{S}'$, we compute the elevation angle $\boldsymbol{e}^s_n$ and compare it with the elevation angle of the light ray $\boldsymbol{e}^l$ to determine potential light occlusion:

$$\boldsymbol{e}^d_n = \boldsymbol{e}^s_n - \boldsymbol{e}^l, \quad \boldsymbol{e}^s_n = \arcsin\left(\frac{\boldsymbol{v}^s_n \cdot \boldsymbol{n}}{\|\boldsymbol{v}^s_n\|_2}\right), \quad \boldsymbol{e}^l = \arcsin\left(\frac{\boldsymbol{v}^l \cdot \boldsymbol{n}}{\|\boldsymbol{v}^l\|_2}\right), \tag{7}$$

where $\boldsymbol{v}^s_n = \boldsymbol{S}'_n - \boldsymbol{p}$ is the vector from point $\boldsymbol{p}$ to surface point $\boldsymbol{S}'_n$, $\boldsymbol{v}^l = \boldsymbol{l} - \boldsymbol{p}$ is the vector from point $\boldsymbol{p}$ to light point $\boldsymbol{l}$, and $\boldsymbol{n} = \begin{bmatrix} 0 & 0 & 1 \end{bmatrix}^\top$ is the normal vector of camera plane. Here, · represents the dot product. The elevation angle is measured relative to the camera plane and lies within the range $(-\pi/2, \pi/2)$. Consequently, the LGI values $\boldsymbol{e}^d$ are naturally bounded within $(-\pi, \pi)$, which is favorable for deep network input stability.

**LGI Maps Generation.** Since the depth map provides only 2.5D information, it cannot capture occluded regions behind foreground objects. An example of ambiguous shadows is shown in Fig. 3d. If the 2.5D representation sufficiently captures the scene geometry (e.g., piece-like objects), a hard shadow mask (Fig. 4e) can be derived by checking whether surface elevation matches light elevation using the condition $\min \left| \boldsymbol{e}^d \right| < \eta$, where $\eta$ is a small threshold (set to $5°$ in our visualization).

In cases where geometric information is insufficient, we defer occlusion reasoning to the model by embedding three-channel LGI maps derived from the elevation difference: ($\boldsymbol{c}^m_1$) the minimum elevation difference, indicating the potential start of occlusion; ($\boldsymbol{c}^m_2$) the maximum elevation difference, indicating the potential end of occlusion; and ($\boldsymbol{c}^m_3$) the value with the smallest absolute difference, representing the most likely point of direct occlusion:

$$\boldsymbol{c}^m_1 = \min_{n=1}^N \boldsymbol{e}^d_n, \quad \boldsymbol{c}^m_2 = \max_{n=1}^N \boldsymbol{e}^d_n, \quad \boldsymbol{c}^m_3 = \boldsymbol{e}^d_{i^\star}, \quad i^\star = \arg\min_{1 \leq n \leq N} \left| \boldsymbol{e}^d_n \right|. \tag{8}$$

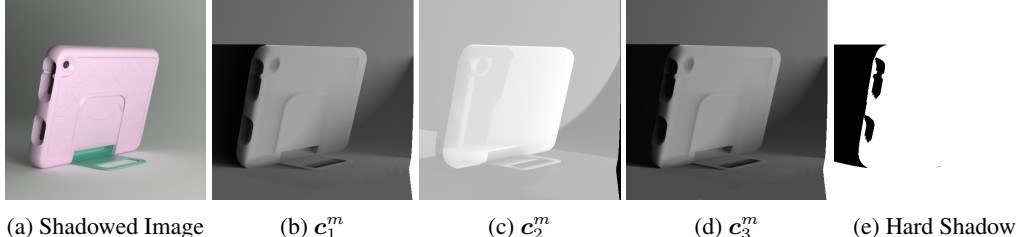

(a) Shadowed Image     (b) $c_1^m$     (c) $c_2^m$     (d) $c_3^m$     (e) Hard Shadow

Figure 4: Example of LGI maps. They indicate shadows cast in the environment and also reveal self-shadowing and shading effects, which can be useful for relighting.

An example visualization of these embeddings is shown in Fig. 4. They capture not only the cast shadows within the environment but also reveal self-shadowing and shading effects. We use the LGI maps ($c^m$) as conditioning signals by concatenating them with image features to guide the bridge-matching in Eq. 3, providing light-aware occlusion cues.

## 4.2 EXTENSION TO IMAGE HARMONIZATION

In addition to explicit light condition setting, we also extend our method to the task of image harmonization, where the light sources are implied by the image, demonstrating its robustness and generalization to outdoor and realistic images. The core architecture of our method remains unchanged. We introduce an additional light estimation network to infer lighting conditions directly from the composited image, removing the need for explicit lighting input. Since our LGI maps are fully differentiable, they enable self-supervised light estimation. Specifically, we use the shadow mask to supervise the predicted lighting. Further details are provided in the supplementary material.

## 4.3 SHADREL DATESET

As no dataset currently exists to support training for coupled light transport, we construct the first large-scale dataset consisting of 817K virtual 3D objects, all crafted by professional 3D artists. The textures of these 3D assets assume the physically-accurate principled Bidirectional Scattering Distribution Function (BSDF) formulation by Burley (2012), capable of simulating a variety of materials including those of glossy, metallic, or transparent appearances. This dataset not only enables effective training but also provides a benchmark for evaluating future methods, and research in coupled illumination and shadow modeling.

To simulate real-world lighting and shadow effects, we render images in Blender using a photorealistic path tracer Cycles. For each object, we generate following types of images for training:

- **Input images**: the object lit by a random High Dynamic Range Imaging (HDRI) environment map drawn from a dataset of panoramas collected from the Internet.

- **Background images**: a planar floor and/or vertical wall of random RGB color, illuminated by a single target point light, where the object is to be inserted.

- **Target images**: the composite of object and background under the same point light — this is the supervision signal for output image. Additionally, a corresponding depth map is also rendered for generating light-occlusion maps.

Each object is augmented by sampling 4 random camera poses and focal lengths; for every camera we further sample 5 point-light-background configurations, totaling 20 target images per object. Point lights vary in position, color, and radius, producing shadows of diverse direction and softness.

Above images are stored in physical units of scene radiance. During training, we apply real-world camera response functions provided by colour-science package, and tone-map results to the $[0, 1]$ standard intensity range. The reader is referred to supplementary material for details about dataset composition, rendering configuration and costs.

## 5 EXPERIMENTS

We evaluate our approach on three benchmark datasets for light-controllable generation and image harmonization (implicit lighting), and present qualitative visualizations on real-world images to demonstrate generalization capability. We further conduct ablation studies, including component analysis, depth-estimation variants, and efficiency evaluation, to validate our design choices.

**Datasets and Evaluation Metrics.** Our method jointly addresses controllable shadow generation and image relighting. As no published dataset supports this combined task, we build a new ShadRel dataset for training and primary evaluation. We also assess the controllable shadow generation component on the public benchmark of Tasar et al. (2024), which targets clean-background shadow generation. For the joint task, we compare with LBM Chadebec et al. (2025), measuring overall image quality (RMSE, SSIM), shadow quality (overall: BER/IoU, shadow: RMSE/SSIM/BER), and object relighting quality (object RMSE/SSIM). We further provide qualitative comparison with SwitchLight Kim et al. (2024). For controllable shadow generation, we compare against CSG Tasar et al. (2024) on their benchmark and metrics (IoU and RMSE). Finally, we extend our method to image harmonization, following Liu et al. (2024b) on the DESOBAv2 dataset Liu et al. (2023), reporting global/local RMSE (GR, LR), SSIM (GS, LS), and BER (GB, LB).

**Implementation Details**. We initialize our framework from Stable Diffusion XL v1.0 Podell et al. (2023). All experiments use $512 \times 512$ images with a batch size of 5. Training employs AdamW optimizer Loshchilov & Hutter (2017) with a learning rate of $3 \times 10^{-5}$. The threshold $\tau$ in Eq. 5 is empirically set to 0.01, and a dilation operation with a kernel size of 17 is applied. In the final loss function, we set the objective weighting factor $\lambda = 10$. The number of sample points $N$ is set to 16. All models are implemented in PyTorch and trained end-to-end.

### 5.1 EXPERIMENTAL RESULTS

**Evaluation on joint shadow synthesis and relighting**. Tab. 1 presents quantitative results on the ShadRel dataset. For fairness, we retrain LBM with the same settings. Our method outperforms LBM in controllable shadow generation and object relighting, demonstrating the effectiveness of our light-geometry interaction formulation. Fig. 5 shows qualitative comparisons, where our method responds more accurately to light and preserves object geometry, producing realistic shadows, precise relighting, and capturing complex object–environment light transport.

Table 1: Joint shadow synthesis and relighting results on ShadRel dataset.

| Method | Overall | | | | Shadow region | | | Object region | |
|---|---|---|---|---|---|---|---|---|---|
| | RMSE ↓ | SSIM ↑ | BER ↓ | IoU ↑ | RMSE ↓ | SSIM ↑ | BER ↓ | RMSE ↓ | SSIM ↑ |
| LBM | 0.0417 | 0.7148 | 0.0847 | 0.7166 | 0.1543 | 0.5690 | 0.1549 | 0.0298 | 0.6797 |
| Ours | **0.0334** | **0.7227** | **0.0588** | **0.8096** | **0.0898** | **0.6195** | **0.1103** | **0.0282** | **0.6875** |

Composite     LBM     Ours     GT       Composite     LBM     Ours     GT

Figure 5: Qualitative comparison with LBM on synthetic object images. Our method produces shadows and relighting effects consistent with geometry and lighting conditions.

**Qualitative analysis on real images**. We further provide qualitative comparisons with SwitchLight and LBM on real-word object images in Fig. 6. SwitchLight is designed for human relighting with point light sources and does not support shadow generation. We compare their method with ours on both human relighting and other real-world objects. Our method produces realistic relighting effects, including for humans, while faithfully adhering to the light direction and preserving the original object colors. In addition, it generates more accurate shadows that align with both the light direction and object geometry, avoiding floating artifacts or inconsistencies in shadow orientation. Despite being trained solely on synthetic data, without any real-world or human-specific samples, our method generalizes well to to real images, including human portraits and complex objects.

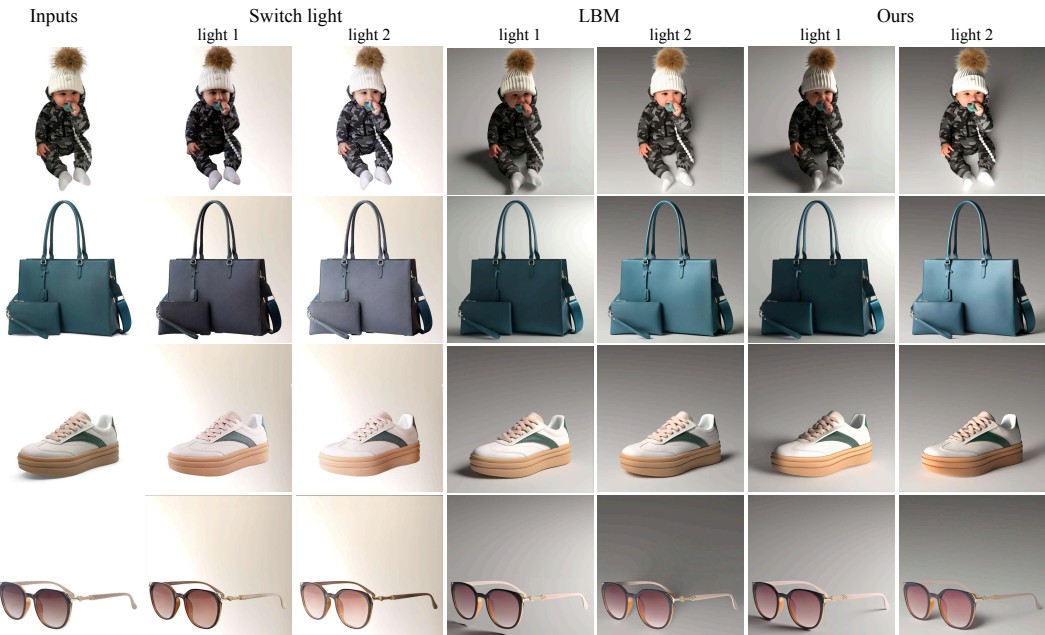

Figure 6: Qualitative comparison with SwitchLight (object relighting without shadow generation) and LBM on real object insertions. Despite being trained solely on synthetic object images, our method outperforms SOTA relighting approaches on real-world object images.

**Evaluation on clean-background shadow generation**. Tab. 2 reports quantitative comparisons with CSG Tasar et al. (2024) on their benchmark. As CSG did not release training data or model weights, we adapt our approach to their clean-background setting by training on ShadRel with pure white backgrounds. Results show that our light–geometry interaction formulation enables more accurate control of shadow shapes. Qualitative results in Fig. 7 further confirm that our method produces shadows with accurate shapes and realistic density.

Table 2: Comparison with CSG Tasar et al. (2024) on their benchmark.

| Method | Track 1 (Softness Control) | | Track 2 (Horz. D. Control) | | Track 3 (Vert. D. Control) | |
|---|---|---|---|---|---|---|
| | IoU ↑ | RMSE ↓ | IoU ↑ | RMSE ↓ | IoU ↑ | RMSE ↓ |
| CSG | 0.818 | **0.021** | 0.780 | **0.030** | 0.776 | 0.028 |
| Ours | **0.821** | **0.021** | **0.798** | **0.030** | **0.785** | **0.027** |

Figure 7: Visual example of clean-background shadow generation, demonstrating that our light-geometry interaction approach provides accurate control over shadow shapes.

**Generalization to Harmonization**. We compare our method with SOTA approaches on the DES-OBAv2 Liu et al. (2023), with quantitative results in Tab. 3 and qualitative comparison in Fig. 8. Our method delivers overall performance comparable to the top-performing approach, SGDGP, while achieving higher accuracy in shadow regions. Although SGDGP also leverages geometry priors, it restricts them to 2D rotated bounding boxes and shadow templates. Qualitative comparisons reveal

Table 3: Comparison on Image Harmonization on DESOBAv2 Dataset.

| Method | BOS Test Images | | | | | | BOS-free Test Images | | | | | |
|---|---|---|---|---|---|---|---|---|---|---|---|---|
| | GR ↓ | LR ↓ | GS ↑ | LS ↑ | GB ↓ | LB ↓ | GR ↓ | LR ↓ | GS ↑ | LS ↑ | GB ↓ | LB ↓ |
| ShadowGAN 45 | 7.511 | 67.464 | 0.961 | 0.197 | 0.446 | 0.890 | 17.325 | 76.508 | 0.901 | 0.060 | 0.425 | 0.842 |
| Mask-SG 12 | 8.997 | 79.418 | 0.951 | 0.180 | 0.500 | 1.000 | 19.338 | 94.327 | 0.906 | 0.044 | 0.500 | 1.000 |
| AR-SG 19 | 7.335 | 58.037 | 0.961 | 0.241 | 0.383 | 0.761 | 16.067 | 63.713 | 0.908 | 0.104 | 0.349 | 0.682 |
| SGRNet 11 | 7.184 | 68.255 | 0.964 | 0.206 | 0.301 | 0.596 | 15.596 | 60.350 | 0.909 | 0.100 | 0.271 | 0.534 |
| DMASNet 32 | 8.256 | 59.380 | 0.961 | 0.228 | 0.276 | 0.547 | 18.725 | 86.694 | 0.913 | 0.055 | 0.297 | 0.574 |
| SGDiffusion 22 | 6.098 | 53.611 | **0.971** | 0.370 | 0.245 | 0.487 | 15.110 | 55.874 | 0.913 | 0.117 | 0.233 | 0.452 |
| SGDGP 47 | **5.896** | 46.713 | 0.966 | 0.374 | **0.213** | 0.423 | 13.809 | 55.616 | **0.917** | 0.166 | **0.197** | 0.384 |
| Ours | 5.900 | **44.753** | 0.961 | **0.415** | 0.239 | **0.415** | **12.979** | **52.543** | 0.912 | **0.201** | 0.213 | **0.358** |

that the main challenge lies in generating shadows that align with object geometry and light direction. Our method addresses this challenge more effectively, as the LGI-based framework directly embeds light–geometry interactions in a space-aligned representation, providing stronger geometric and illumination cues. This leads to improved robustness in outdoor and realistic scenarios.

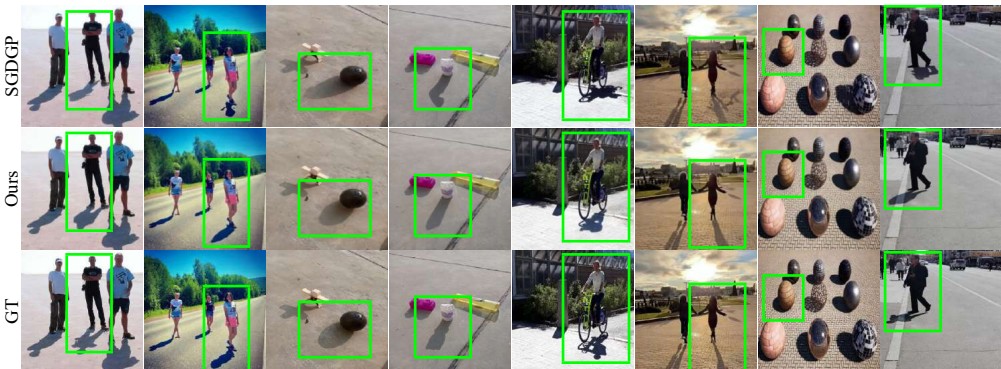

Figure 8: Qualitative comparison with SGDGP Zhao et al. (2025) . Our method achieves higher accuracy in shadow regions by aligning more closely with object geometry and light direction.

## 5.2 ABLATION STUDY

Table 4: Analysis of Method Components on ShadRel Dataset

| Components | | Overall | | | | Shadow region | | | Object region | |
|---|---|---|---|---|---|---|---|---|---|---|
| W-L1 | LGI | RMSE ↓ | SSIM ↑ | BER ↓ | IOU ↑ | RMSE ↓ | SSIM ↑ | BER ↓ | RMSE ↓ | SSIM ↑ |
| | | 0.0408 | 0.7109 | 0.1012 | 0.7193 | 0.1236 | 0.5868 | 0.1923 | 0.0330 | 0.6679 |
| ✓ | | 0.0391 | 0.7148 | 0.0940 | 0.7353 | 0.1154 | 0.5974 | 0.1784 | 0.0317 | 0.6680 |
| ✓ | ✓ | 0.0334 | 0.7227 | 0.0588 | 0.8096 | 0.0898 | 0.6195 | 0.1103 | 0.0282 | 0.6875 |
| -LGI+Depth | | 0.0388 | 0.7148 | 0.0932 | 0.7344 | 0.1166 | 0.5938 | 0.1765 | 0.0315 | 0.6719 |
| +Depth | | 0.0339 | 0.7188 | 0.0719 | 0.7921 | 0.0942 | 0.6139 | 0.1364 | 0.0283 | 0.6836 |
| LGI Ch3 | | 0.0351 | 0.7179 | 0.0670 | 0.7824 | 0.0932 | 0.6095 | 0.1343 | 0.0290 | 0.6807 |
| w/ DAv2 | | 0.0334 | 0.7188 | 0.0602 | 0.8148 | 0.0901 | 0.6212 | 0.1269 | 0.0283 | 0.6875 |
| w/ GT depth | | 0.0326 | 0.7266 | 0.0558 | 0.8107 | 0.0894 | 0.6195 | 0.1070 | 0.0280 | 0.6875 |

We analyze component contributions in Tab. 4, using LBM with standard L1 loss as the baseline. Each added module yields consistent performance gains [1]. Comparing LGI embeddings with direct depth embeddings shows that depth alone ('-LGI+Depth') brings only marginal improvements, while combining both ('+Depth') slightly degrades performance, likely due to noise and misalignment in depth estimation. Using only the third channel of LGI ('LGI Ch3') also reduces perfor-

---

[1] A visual comparison with and without the LGI module is provided in the Appendix G.

mance, confirming the necessity of proposed channels. Finally, replacing the original depth estimator with DepthAnythingV2 Yang et al. (2024) ('w/ DAv2') and ground truth depth produces minimal variation, demonstrating robustness to the choice of depth backbone.

Our model requires $4.82\,\mathrm{GB}$ of parameter memory and $3.12\,\mathrm{TFLOPs}$, increasing these costs over the baseline by only 0.0004% and 0.0011%, respectively, underscoring its efficiency. The method further extends naturally to multiple objects and light sources, with additional details, visualizations, and an analysis of the impact of the sample-point count $N$ provided in the Appendix D.

## 6 CONCLUSION

We presented a unified, physically inspired framework utilizing Light-Geometry Interaction maps for joint shadow generation and object-level relighting. By approximating scene geometry from predicted depth maps, our method generates coherent and physically plausible shadows and relighting effects without requiring full 3D reconstruction. Additionally, we introduced a high-quality synthetic dataset specifically designed for joint shadow generation and relighting tasks, featuring diverse object categories, multiple material types, and challenging scenarios involving object-environment interreflection. Extensive experiments confirm that our approach achieves SOTA performance, effectively balancing visual consistency, physical accuracy, efficiency, and generalizability.

## ACKNOWLEDGMENTS

This work was conducted during an internship at Amazon. Hongdong Li is co-employed by Australian National University (full-time, primary) and Amazon IML (as a part-time Amazon Scholar). Hongdong Li is also partially supported by an ARC Discovery Grant DP220100800.

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

## A    EXTEND TO SHADOW-CENTERED IMAGE HARMONIZATION

In this section, we extend our method to realistic image harmonization, where a foreground object is composited onto a new background. Most harmonization datasets do not provide ground-truth lighting information; however, some datasets include shadow masks. In our setting, the model infers lighting conditions directly from the composited image. As shown in Fig. 9, the light estimation network predicts a complete set of lighting parameters, including color, radius, distance, intensity, and direction. These parameters are integrated into our light-aware shadow generation and relighting framework to ensure consistent and realistic harmonization of the inserted object. Since LGI is fully differentiable, it enables end-to-end training of the light estimation network. Furthermore, when shadow mask is available, we formulate a loss function that leverages LGI to supervise the estimation of lighting direction.

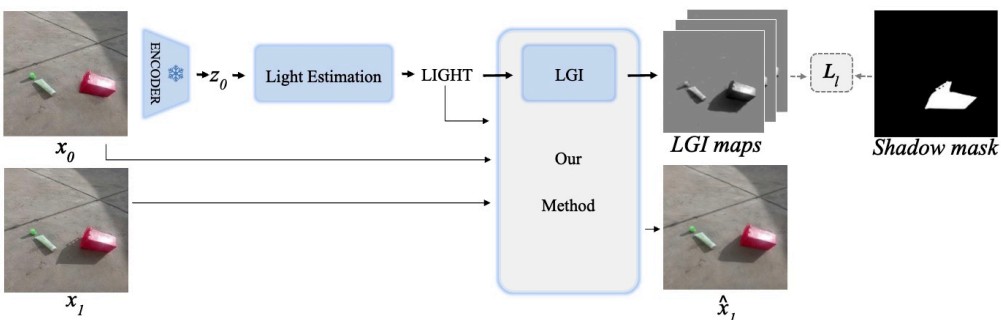

Figure 9: Overview of the image harmonization pipeline. The light estimation network predicts a full set of lighting parameters, with the estimated lighting direction used to compute LGI maps supervised by ground-truth shadow masks. These estimates are integrated into our light-aware shadow generation and relighting framework to produce consistent and realistic harmonization of the inserted object. Since the LGI module is fully differentiable, it enables end-to-end training of the light estimation network.

The light estimation network consists of four convolutional layers with progressively increasing channel dimensions. The resulting feature representation is then processed by a two-layer multilayer perceptron (MLP), producing an 8-dimensional vector encoding the full set of lighting parameters. During training, we first freeze the light-aware shadow generation and relighting framework for 5 epochs to warm up the light estimation module, and then continue with end-to-end training where all parameters are updated.

For outdoor scenarios, sunlight serves as the primary light source and can be effectively modeled as a point light at an infinite distance, emitting parallel rays. To accommodate this scenario, we introduce a modified variant of our LGI maps specifically adapted for sunlight. Instead of casting rays toward a finite light position, rays are cast along the estimated light direction relative to our normalized camera coordinate system, where the x-axis and y-axis lie within the range $(-0.5, 0.5)$ and the z-axis within $(0, 1]$. According to this coordinate system in Eq. 6, we assume a default intrinsic matrix $K = \begin{bmatrix} W & 0 & W/2 \\ 0 & H & H/2 \\ 0 & 0 & 1 \end{bmatrix}$, where H, W denote the height and width of the input image. An visual example of LGI maps for sunlight is shown in Fig. 10.

Building on the LGI maps, we obtain the predicted hard shadow mask $\mathcal{M}_p$. We then define a light-direction estimation loss $\mathcal{L}_l$, applied when a ground-truth shadow mask $\mathcal{M}_g$ is available. Since we only consider the inserted object's shadow, the loss is computed within a region restricted to the dilated ground-truth shadow mask. The overall loss combines BCE and IoU terms:

$$\mathcal{L}_l(\mathcal{M}_p, \mathcal{M}_g) = \mathcal{L}_{\text{BCE}}(\mathcal{M}_p, \mathcal{M}_g) + \mathcal{L}_{\text{IoU}}(\mathcal{M}_p, \mathcal{M}_g), \tag{9}$$

where the BCE loss is defined as

$$\mathcal{L}_{\text{BCE}}(\mathcal{M}_p, \mathcal{M}_g) = -\mathbb{E}\Big[\mathcal{M}_g \log \mathcal{M}_p + (1 - \mathcal{M}_g) \log(1 - \mathcal{M}_p)\Big], \tag{10}$$

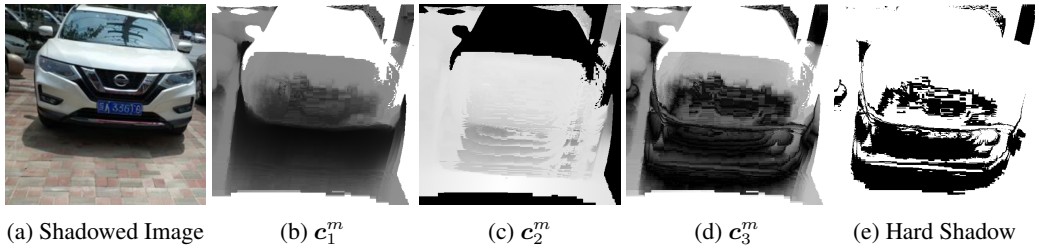

| (a) Shadowed Image | (b) $c_1^m$ | (c) $c_2^m$ | (d) $c_3^m$ | (e) Hard Shadow |

Figure 10: Example of sunlight LGI maps. They establish a correspondence between light direction and shadow shape when scene geometry is known.

and the IoU loss is defined as

$$\mathcal{L}_{\text{IoU}}(\mathcal{M}_p, \mathcal{M}_g) = 1 - \frac{\mathbb{E}[\mathcal{M}_p\mathcal{M}_g]}{\mathbb{E}[\mathcal{M}_p + \mathcal{M}_g - \mathcal{M}_p\mathcal{M}_g]}, \tag{11}$$

with $\mathbb{E}$ denoting the expectation over pixels.

## B  MULTIPLE OBJECTS

Our method extends naturally to scene editing via object-level insertion. We further demonstrate multi-object insertion by applying the method sequentially, inserting objects one at a time. As shown in Fig. 11, our approach generalizes well to the multi-object setting: it casts realistic shadows onto previously inserted objects, ensures that the generated shadows align with the scene geometry, and maintains relighting consistent with the materials of each object.

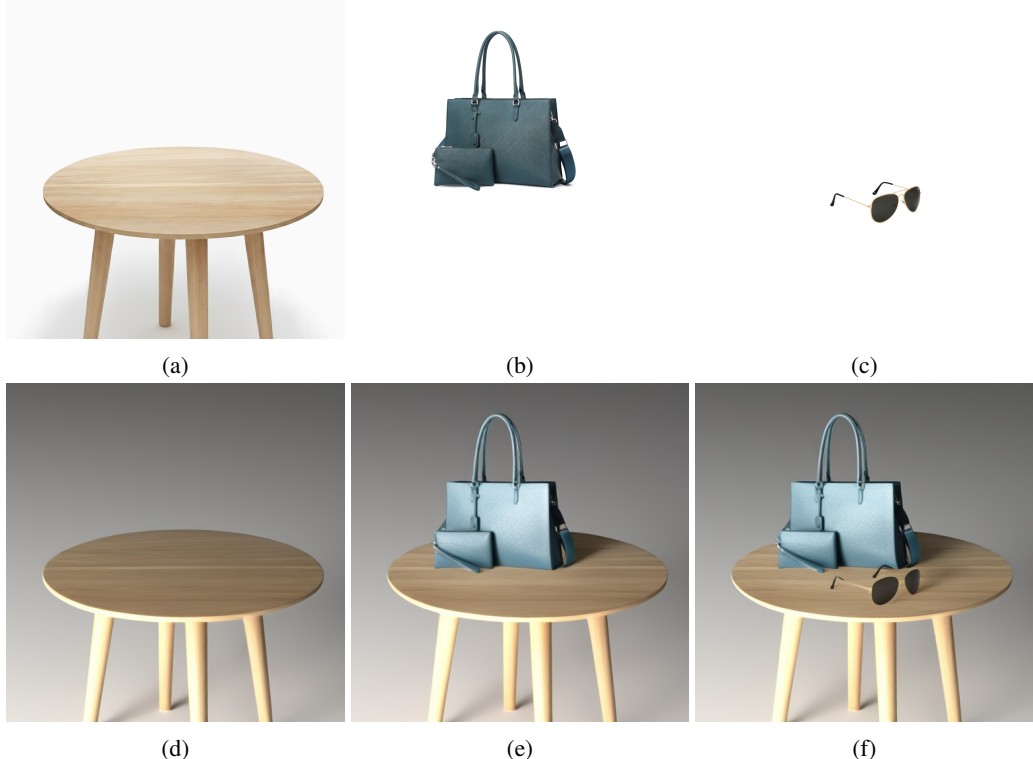

Figure 11: Examples of multiple object insertion. (a–c) Inserted objects. (d–f) Scenes with inserted objects after shadow generation and relighting. Our method produces realistic, texture-aware shadows on the table and preserves faithful relighting across wood, leather, metal, and glass materials.

## C  MULTIPLE LIGHT SOURCES

While our primary focus is on single-light scenarios, the method naturally extends to multiple light sources by exploiting the linearity of radiance  Debevec (2008); Ward (1994).  Specifically, we accumulate per-light contributions through additive composition:

$$x_1 = \sum_{l=1}^{L} x_1^{(l)}, \tag{12}$$

where $x_1^{(l)}$ denotes the relight result under the l-th light, and $L$ is the total number of light sources. This straightforward accumulation requires no additional modification and still captures complex phenomena such as overlapping shadows and varying intensities, as shown in Fig. 12.

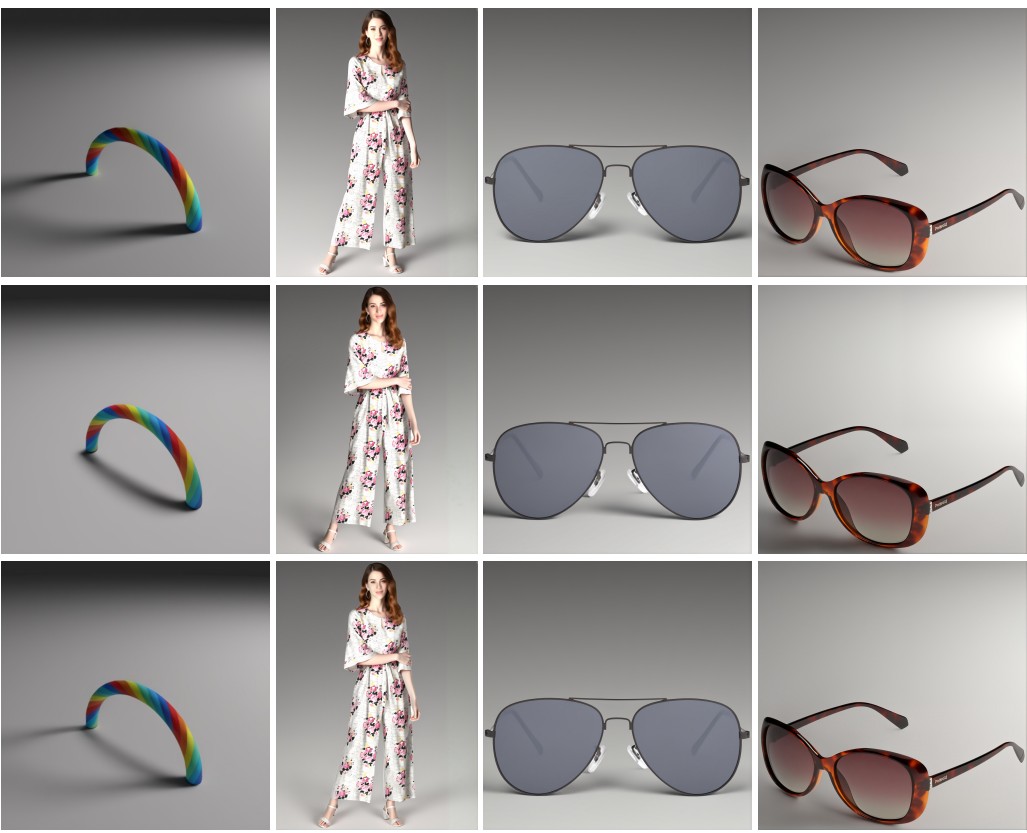

Figure 12: Examples of scenes with various light sources (top–middle) rendered using our method, and fused two-light-source scenes (bottom)

## D  STUDY ON SAMPLE-POINT COUNT

Table 5: Study on Sample Point Count

| N | Overall | | | | Shadow region | | | Object region | |
|---|---|---|---|---|---|---|---|---|---|
|   | RMSE↓ | SSIM↑ | BER↓ | IOU↑ | RMSE↓ | SSIM↑ | BER↓ | RMSE↓ | SSIM↑ |
| 8  | 0.0344 | 0.7188 | 0.0612 | 0.7954 | 0.0991 | 0.6125 | 0.1141 | 0.0283 | 0.6875 |
| 16 | 0.0334 | 0.7227 | 0.0588 | 0.8096 | 0.0898 | 0.6195 | 0.1103 | 0.0282 | 0.6875 |
| 32 | 0.0334 | 0.7231 | 0.0586 | 0.8099 | 0.0869 | 0.6198 | 0.1096 | 0.0283 | 0.6875 |

We further study the effect of the sample-point count $N$. Because its computational impact is negligible at our reporting precision, Tab. 5 reports accuracy for $N \in \{8, 16, 32\}$. While $N{=}32$ provides only marginal gains, we adopt $N{=}16$ by default.

# E    COMPARISON TO LBM WITH DEPTH INPUT

We additionally compare with LBM equipped with depth input in Tab. 6. Our method consistently outperforms this depth-augmented LBM across all regions and metrics, showing that the gains of LGI stem from our proposed design rather than merely from using an additional depth modality.

Table 6: Comparison to LBM with Depth Input

|  | Overall | | | | Shadow region | | | Object region | |
|---|---|---|---|---|---|---|---|---|---|
|  | RMSE↓ | SSIM↑ | BER↓ | IOU↑ | RMSE↓ | SSIM↑ | BER↓ | RMSE↓ | SSIM↑ |
| LBM+depth | 0.0395 | 0.7122 | 0.1003 | 0.7284 | 0.1208 | 0.5923 | 0.1901 | 0.0321 | 0.6692 |
| Ours | 0.0334 | 0.7227 | 0.0588 | 0.8096 | 0.0898 | 0.6195 | 0.1103 | 0.0282 | 0.6875 |

# F    MORE VISUALIZATION UNDER DIFFERENT LIGHTS

In this section, we present additional real-world examples of shadow generation and relighting under varying light sources (Fig. 13). Despite being trained exclusively on synthetic data, our model generalizes effectively to real images and successfully captures indirect lighting effects, such as reflections. More visualizations are included in the supplementary video.

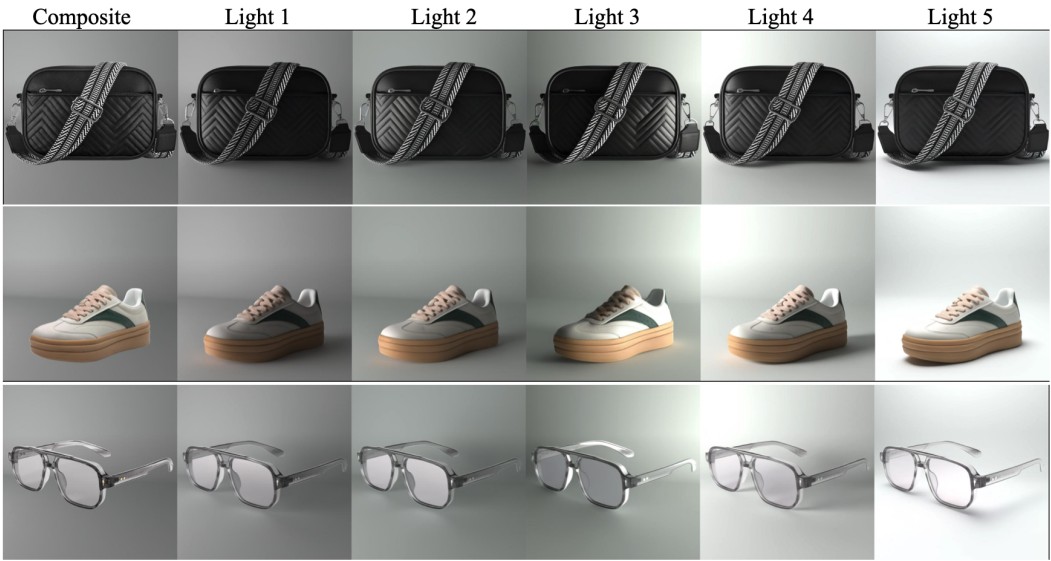

Figure 13: Examples produced by our method on real-world object images.

# G    VISUAL COMPARISON WITH AND WITHOUT THE LGI MODULE

Fig.14 compares results with and without the LGI module. Incorporating LGI increases sensitivity to scene geometry, yielding shadows that are more realistic and better aligned with geometric structure.

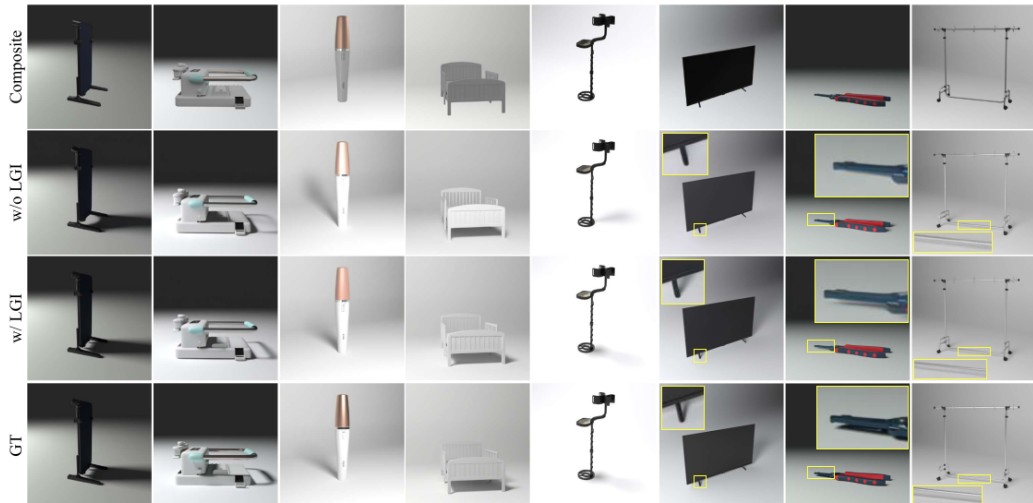

Figure 14: Qualitative comparison with and without LGI, illustrating that LGI enables geometry-aligned shadows and relighting effects.

## H COMPLEX BACKGROUND EXAMPLES

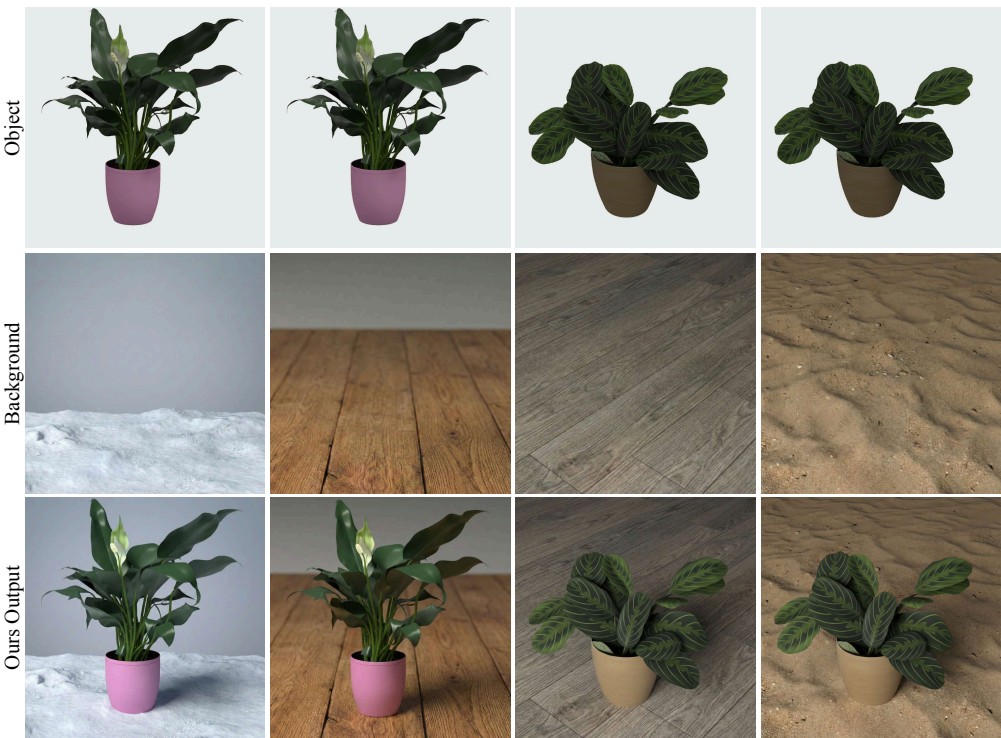

Figure 15: Qualitative results on complex backgrounds. Our method remains robust on both non-planar and textured backgrounds.

In this section, we shift our focus to complex backgrounds. While the proposed ShadRel dataset is primarily designed to study complex object–background inter-reflections and therefore uses relatively simple backgrounds (e.g., a floor or a floor–wall configuration), the objects themselves span a wide range of materials, geometries, and textures. As a result, complex geometry and texture are

still exercised through self-occlusion effects. Our real-image visualizations in Fig. 11 show that the method effectively handles textured backgrounds (e.g., desks, cluttered layouts, and object-rich scenes). Additional qualitative results are provided in Fig. 15, where we observe that our method remains robust on non-planar and textured backgrounds.

Furthermore, the DESOBAv2 dataset is a real-world benchmark with images captured on golf courses, tennis courts, beaches, gardens, and other outdoor environments, where the backgrounds are non-planar and contain rich textures and patterns. The qualitative results on these challenging cases in Fig. 16 demonstrate that our method is robust to such complex backgrounds. We further demonstrate its generalization capabilities on in-the-wild images in Fig. 17, showing that our method generates reasonable, lighting-consistent shadows even on non-planar and textured backgrounds.

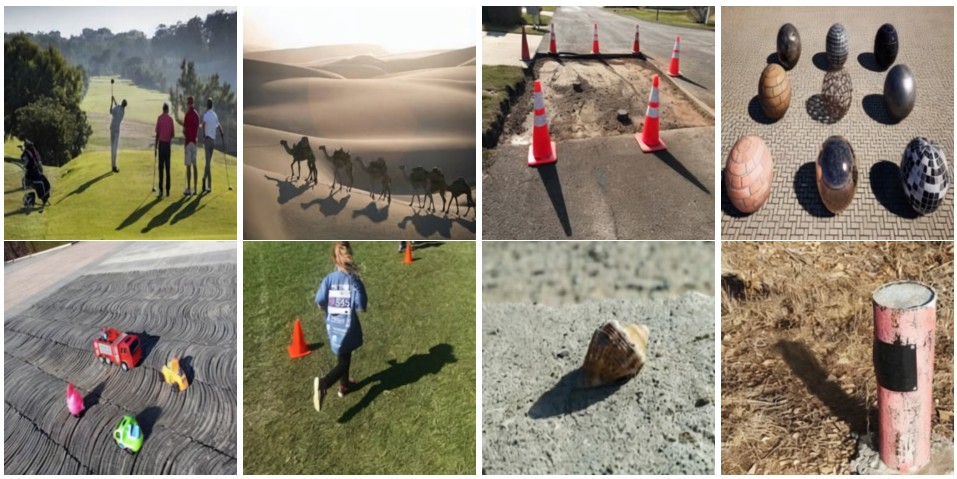

Figure 16: Complex backgrounds in the DESOBAv2 dataset, including non-planar geometry and rich textures and patterns. Our qualitative results on this dataset show that the proposed method produces reasonable predictions even under these challenging background conditions.

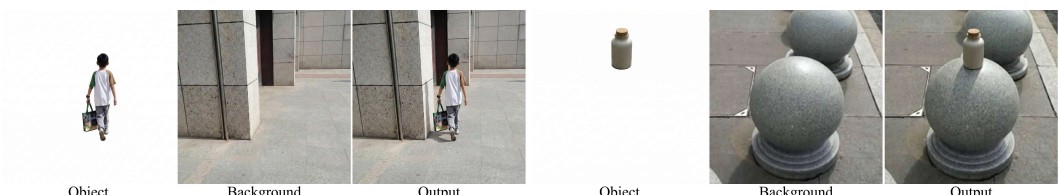

Figure 17: Visual results on challenging in-the-wild backgrounds. Our model maintains lighting consistency and shadow realism across diverse surface geometries and textures.

# I STUDY ON COMPLEX OBJECTS

To quantitatively evaluate our method on complex objects, we construct a 200-sample subset consisting solely of challenging complex objects and evaluate our method on this subset. The quantitative results in Tab. 7 show our method outperforms LBM by a large margin, together with the additional qualitative examples in Fig. 18, demonstrate that our approach remains robust beyond simple shapes. We further provide additional qualitative results on this subset and on real images in the supplementary video. Moreover, the DESOBAv2 dataset also includes complex objects, such as humans with bicycles in Fig. 8; our results on this dataset likewise show that the method preserves high-quality predictions in these cases, further supporting its robustness to complex geometries.

Table 7: Study on Complex Objects

| | Overall | | | | Shadow region | | | Object region | |
|---|---|---|---|---|---|---|---|---|---|
| | RMSE↓ | SSIM↑ | BER↓ | IOU↑ | RMSE↓ | SSIM↑ | BER↓ | RMSE↓ | SSIM↑ |
| LBM | 0.0431 | 0.7112 | 0.0954 | 0.0691 | 0.1575 | 0.5609 | 0.1758 | 0.0342 | 0.6743 |
| Ours | 0.0341 | 0.7196 | 0.0658 | 0.7940 | 0.0929 | 0.6114 | 0.1241 | 0.0286 | 0.6828 |

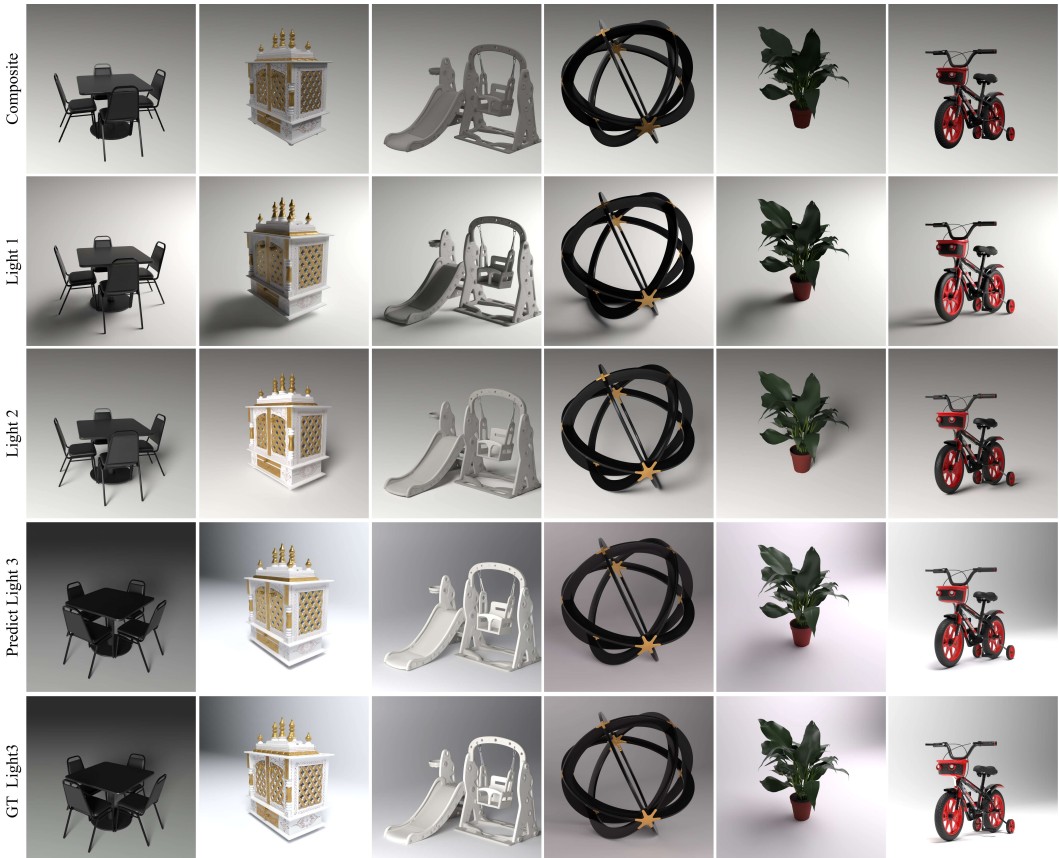

Figure 18: Qualitative results on complex objects. Our ShadRel dataset includes complex objects, and our method remains robust on these challenging cases. As the dataset provides Ground Truth (GT) for only 5 discrete lighting conditions, the Light 3 shows a direct comparison against the available GT. Light 1 and Light 2 represent unseen illumination directions (sampled from the 24 distinct lights in the supplementary video), demonstrating the model's generalization capabilities where GT is unavailable.

## J  VISUALIZATION ON COMPLEX INTERACTION

In this section, we present additional qualitative results on complex object–background interaction effects (Fig. 19), including cast shadows from transparent objects and shadows modulated by inter-reflections and secondary reflections. When inserting an object into a scene, not only is the object relit by the background, but the background is also altered by the object (e.g., through shadows and reflections), and these changes in turn further influence the object. Such higher-order object–background interactions remain an open challenge. Our ShadRel dataset is specifically constructed to capture these challenging phenomena, featuring strong reflections from materials such as glass and weaker reflections from fabrics—scenarios that are close to real applications but rarely covered by existing models and datasets. Even under these demanding conditions, our method

produces high-fidelity, physically consistent predictions that align well with the ground truth, highlighting both the unique difficulty of ShadRel and the effectiveness and robustness of our approach.

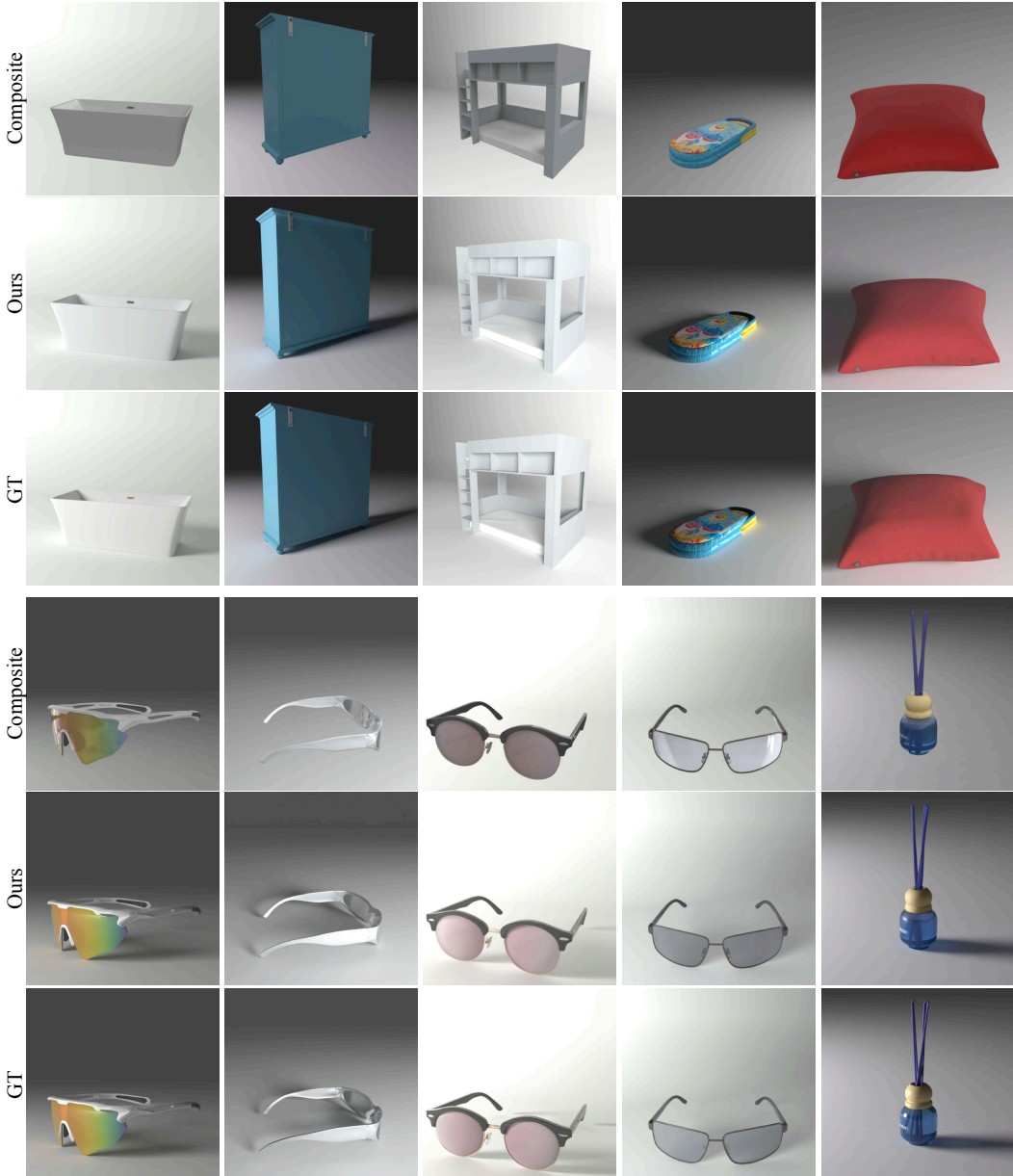

Figure 19: Qualitative results on complex object-background interaction. ShadRel dataset contains objects with diverse materials, including reflective ones such as glass and porcelain, which introduce challenging shadows from transparent objects, with both primary and secondary reflections. Even in these difficult scenarios, our method produces high-fidelity, physically consistent predictions.

## K    DATASET COMPOSITION AND EXAMPLES

In this section, we present example images from our dataset. Each sample contains 4 random camera poses, and for each pose we generate 5 point-light–background configurations. As shown in Fig. 20, the dataset consists of input images of objects, background images rendered under varying lighting conditions (floor and occasionally a wall), and target images where the same objects and backgrounds are illuminated with different lighting effects.

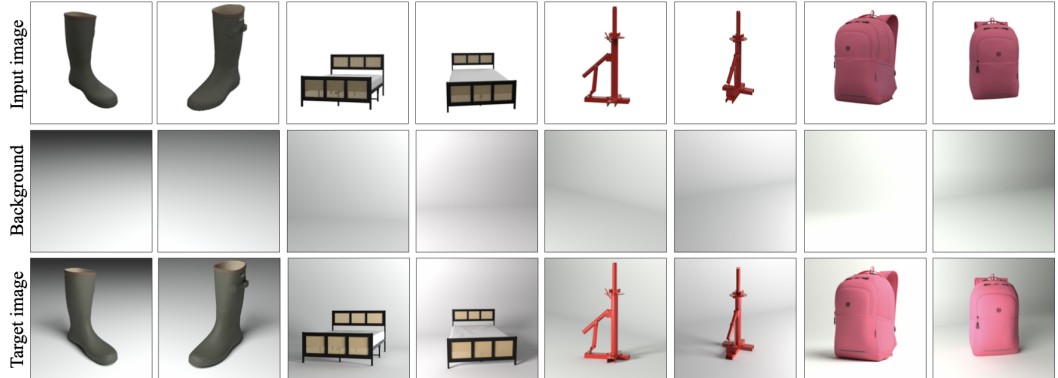

Figure 20: Examples from our dataset. Each sample includes (top) input images of objects from different camera poses, (middle) background images with floor and optionally a wall under point-light illumination, and (bottom) target images combining the objects and backgrounds under the same point light.

## L  LIMITATIONS

Since our approach is initialized from a diffusion model, it inevitably inherits some of the well-known limitations of this class of generative models. Most notably, the generated content may not remain fully faithful to the input specification, particularly in regions requiring high-fidelity reproduction of fine-grained details. As illustrated in Fig. 21, elements such as logos can appear inconsistent, distorted, or incomplete.

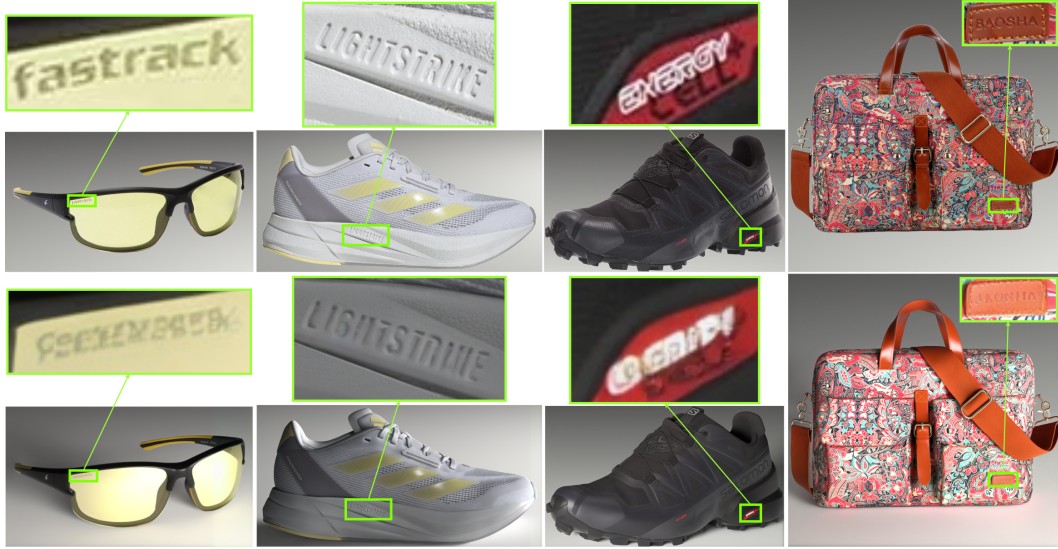

Figure 21: Examples illustrating the limitations of our method in reproducing fine-grained details. (Top: Ground Truth; Bottom: Our results.) While our approach can handle certain structural details, tiny elements, such as Logos, may appear inconsistent, distorted, or incomplete.

