# OpenReview forum: "Joint Shadow Generation and Relighting via Light-Geometry Interaction Maps"
_ICLR.cc/2026/Conference — ICLR 2026 Poster_

### Official Review · Reviewer_2xqw · 2025-10-29

**Soundness:** 3
**Presentation:** 3
**Contribution:** 2
**Rating:** 2
**Confidence:** 3

**Summary:**

This paper proposes a framework to generate shadow maps according to the specified lighting condition and supports object-background composition with plausible shadows and relighting. The paper leverages latent bridge mapping to generate the results, and condition the model by introducing Light–Geometry Interaction (LGI) maps. The LGI map leverages information from the 2.5D estimated depth map and captures light-geometry interaction by the elevation angle of the reprojected points from the depth map. The LGI provides light-aware occlusion cues and controls the diffusion model to generate plausible results. Experiments show plausible object insertion results with realistic shadow effects.

**Strengths:**

1. Reasonable design. The idea of using 2.5D information from depth maps to obtain occlusion cues is reasonable. The proposed LGI method leverages ray sampling and reprojection, which constitute a physically informed algorithm.
2. Dataset contribution. The paper proposes a large-scale dataset, ShadRel, providing synthetic data for lighting and shadow effects. The research community can benefit from the proposed dataset for training and benchmarking.

**Weaknesses:**

## Major
1. Limited novelty. This paper is built upon the existing LBM framework with a new LGI representation, which is the paper's main contribution, as a control signal providing occlusion cues. However, the idea of ray sampling in LGI is not new -- similar to screen-space raytracing (SSRT) in computer graphics and existing works [1,2] already use this for object insertion, cast shadows and relighting.
2. Simple setting. Most experiments use the setting of inserting the object into a planar background with point light sources, which is a simple setting and not very interesting. The image harmonization setting is more interesting -- but the paper only provides limited results in Fig. 7.
3. Missing baselines. Although this paper is more focused on shadow generation, general-purpose image-based relighting/harmonization methods such as IC-Light [3] are still important baselines since they also handle shadows, especially for the image harmonization setting.

## Minor
1. Low image quality in the figures. The paper uses bitmap images for Figs. 2, 6, 7, causing sawtooth artifacts in fonts when zooming in. In addition, the image resolution in Fig. 7 is low. I may expect a higher-quality result for image harmonization.
2. Missing citations. Lots of relevant papers, such as SSRT-related graphics papers [1,2] or image-based lighting papers [3] are not cited.

## References
- [1] Griffiths, David, Tobias Ritschel, and Julien Philip. "OutCast: Outdoor Single‐image Relighting with Cast Shadows." Computer Graphics Forum. Vol. 41. No. 2. 2022.
- [2] Zhu, Jingsen, et al. "Learning-based inverse rendering of complex indoor scenes with differentiable monte carlo raytracing." SIGGRAPH Asia 2022 Conference Papers. 2022.
- [3] Zhang, Lvmin, Anyi Rao, and Maneesh Agrawala. "Scaling in-the-wild training for diffusion-based illumination harmonization and editing by imposing consistent light transport." The Thirteenth International Conference on Learning Representations. 2025.

**Questions:**

1. Can the authors clarify the difference between the proposed LGI and SSRT in the rebuttal and summarize their contributions in this design?
2. Can the proposed method work well on object insertion into more complex background environments, rather than a simple planar background with point light sources?
3. The example illustrated in Fig. 3 is interesting, but I don't find how the proposed method handles the ambiguous case in Fig. 3(d). How will the LGI encode the "not sure" information in such case?

---

> ### Author Response · Authors · 2025-11-25
>
> We sincerely thank the reviewer for recognizing the method design and dataset contribution, as well as for the valid concerns. We address each concern below and have incorporated the corresponding revisions (highlighted in blue) in Sec. 2, App. J, and App. L of the paper.
>
> ---
>
>  ## Novelty
> Our contribution is not the ray-sampling strategy but the LGI representation. Instead of using ray sampling to collect RGB/features for shadow or radiance regression as in [1,2], LGI encodes only light–scene geometry interactions as a differentiable, elevation-based occlusion cue that guides a generative model toward physically consistent relighting and shadow generation. Its formation is much simpler than the feature aggregation approach of [1,2], derived from only scene depth and light pose. More specifically, we further note the following differences:
>
> [1] uses ray sampling to collect RGB and depth values for shadow regression, which requires ground-truth shadow supervision, whereas LGI does not.
>
> [2] uses SSRT (ray sampling) to extract features such as normals, albedo, and roughness to train a lighting network that regresses incident radiance rather than occlusion cues, and it requires ground-truth normals, albedo, and roughness for supervision.
>
> Thus, although ray sampling is used as a standard sampling approach, the resulting representations, supervision targets, and their functional roles differ fundamentally.
>
> ---
>
> ## Simple setting
>
> Our method can be extended to harmonization under more diverse settings. As shown in Fig.16 and Fig.17, where our method shows good generalization to complex backgrounds. Additionally,  App. C shows that our method extends naturally to multiple lights via radiance linearity. Furthermore, unlike conventional object insertion methods, ours is the first capable of not only relighting foreground objects but also producing foreground-background lighting interactions such as shadows, inter-reflections, and refractions from two images (please refer to Fig.19). Handling two-way light transportation is an arguably more challenging problem setting for many downstream applications.
>
> For the harmonization setting, we also provide quantitative results in Table 3. Our evaluations already cover datasets with substantial background complexity, such as non-planar geometry and rich textures in DESOBAv2 (see Fig. 17). Our strong performance on DESOBAv2 confirms that the method can robustly handle such complexity in practice.
> Despite the simpler backgrounds in ShadRel, joint shadow generation and object relighting remain challenging for objects with complex geometry and texture due to self-shadowing on intricate surfaces. Thus, our model is also optimized to handle such cases. As shown in Fig. 1 and Fig. 16, it produces high-quality results even with non-planar, textured, and cluttered backgrounds.
>
>
> ---
>
> ## Missing baselines
>
> IC-Light and our method have different emphases. IC-Light is designed mainly for foreground object relighting from a given environment map. In contrast, our method focuses on producing physically consistent object–background light interactions, including not only foreground relighting but also recovering shadows and inter-reflections between the foreground and background.
>
> Because of this difference, the DESOBAv2 dataset is not an appropriate benchmark for IC-Light. DESOBAv2 is a pure shadow-generation dataset: the foreground appearance is fixed, and only the shadow needs to be synthesized. This creates a significant domain gap for IC-Light, whose strength lies in adjusting foreground object appearance rather than generating shadows. In our qualitative attempts to run IC-Light on DESOBAv2 examples as shown in [link](https://github.com/iclranonymous2026-vis/ICLR_2026/blob/main/IClight_DESOBAv2/README.md), we observed that it often ignores or only partially reproduces shadows and incorrectly relights the object, confirming a mismatch between the method’s design and the dataset’s supervision target.
> Furthermore, fine-tuning IC-Light on DESOBAv2 is not feasible. IC-Light requires training triplets captured under $light_1$​, $light_2$​​, and $light_{1+2}$​​. However, DESOBAv2 provides no paired scenes under varying illumination, making proper supervision impossible.
> Thus, although IC-Light is a strong relighting method, it is not a meaningful or fair baseline for a shadow-generation task or joint shadow generation and relighting.
>
> ---
>
> ## Low image quality
> We appreciate the comment and have modified it in the revised version.
>
> ---
>
> ## Citations
> We originally discussed [1] (lines 105–119) and [3] (line 110) in the related work section. We have now added [2] and expanded our discussion of these works.
>
> ---
>
> ## Ambiguous case in Fig. 3(d)
> We do not explicitly encode a “not sure’’ state in LGI. Ambiguous regions are handled implicitly by the diffusion model.

---

> > ### Comment · Reviewer_2xqw · 2025-11-27
> >
> > I appreciate the authors' response and the updated results in the revision. However, I still have several issues as below:
> > 1. Novelty: I can understand that this work and [1,2] use the ray-sampling (or SSRT) strategy for completely different purposes, but I still suggest the authors should tone down the description of this part's contribution due to the similar underlying algorithm. Moreover, I believe that the SSRT algorithm itself in [1,2] also only relies on the depth map itself, while other features (e.g., albedo, normal, etc.) are used for their task-specific network input rather than ray tracing. Therefore, I believe that [1,2]'s SSRT algorithm is very similar to this paper's ray sampling strategy in principle, although for completely different purposes.
> > 2. Complex objects: I appreciate the updated result in Fig. 15 and acknowledge that the results look plausible. However, since they're all synthetic data, could you also provide the ground truth images for a qualitative comparison?
> > 3. Image harmonization settings: I appreciate the updated results and acknowledge that this paper's method achieves plausible relighting and shadow effects for both the foreground object and the background environment. I noticed that in all the harmonization results the shadow in the background has a "planar" pattern -- the background is considered as a horizontal plane on which the shadow is cast. I'd like to know if the method is able to handle a more complex "non-planar background" setting, for example, an object inserted near the corner of a room so that the shadow of the lower part of the object is cast on the planar floor while the shadow of the upper part is cast on the vertical wall, or even a more challenging scenario where the object's shadow is cast onto a curved surface. If the method can handle such cases, this will largely validate its applicability to in-the-wild scenes; if not, it should be discussed as a limitation/failure case.
> > 4. Fig. 3(d)'s ambiguous case: If my understanding is correct, according to the criteria defined by Eq. 7 and L258 ($\min |\mathbf{e}^d| < \eta$, the Fig. 3(d)'s case will be considered as non-occlusion, since $\mathbf{e}^l$ and $\mathbf{e}^s_n$ have different signs (different sides of the camera plane) so that $|\mathbf{e}^d| = |\mathbf{e}^l| + |\mathbf{e}^s_n|$ is likely to be large and will not meet the criteria. Is my understanding correct? If so, will this cause an underestimation of the ambiguous cases since they'll always be considered as non-occlusion?

---

> > > ### Author Response · Authors · 2025-11-30
> > >
> > > We sincerely thank the reviewer for the constructive follow-up and detailed engagement with our work. We appreciate the acknowledgment of our updated results, and we address the remaining concerns below and have incorporated the corresponding revisions (highlighted in magenta) in App. I and App. J of the paper.
> > >
> > > ---
> > >
> > > ## Novelty
> > > We thank the reviewer for the opportunity to clarify this. First, we would like to highlight that **we had never listed the ray-sampling as a contribution in the original submission**. The submitted original manuscript focuses on the LGI representation, which is the core contribution of the work. As the LGI representation is derived from the ray sampling technique, we had included some descriptions of ray sampling in Line 231-237 for the completeness of our presented method.
> > >
> > > On the other hand, although both [1, 2] and our proposed method adopt the depth information, its utilization serves fundamentally different purposes, which are detailed as follows.
> > >
> > > **[1] uses depth as a "Feature Channel"**: It does not explicitly explore the geometric topology; rather, it inputs depth combined with RGB as concatenated channel information for regression.
> > >
> > > **[2] uses depth as a "Stop Signal"**: The ray-tracer stops at surface intersections determined by the depth map. Their goal is modeling surface radiance, not occlusion.
> > >
> > > **Ours uses depth as the "Geometric Signal"**: We transform the depth topology itself into a relative elevation metric to compute occlusion.
> > >
> > > ---
> > >
> > > ## Ground Truth for Complex Objects
> > > We thank the reviewer for the positive assessment of our results. Regarding the ground truth (GT) comparison, we wish to clarify that our synthetic dataset provides GT images for only 5 discrete lighting conditions.
> > > The original visualization in Fig. 15 and the supplementary video utilize 24 distinct lighting directions to demonstrate the generalization of our model to unseen illumination; consequently, ground truth does not exist for these specific frames. To address your request, we have added two rows to Fig.15, which provide direct qualitative comparisons in cases where GT images are available for the corresponding lighting conditions.
> > >
> > > ---
> > >
> > > ##  Non-Planar Backgrounds
> > > We appreciate the reviewer raising this interesting point regarding complex geometries. We are happy to clarify that our method is **not restricted to planar surfaces**. Since our LGI representation is derived from ray-marching on the pixel-wise composite depth map, it naturally generalizes to arbitrary background geometries.
> > >
> > > To demonstrate the capacity of our proposed method in handling non-planar background, we provide additional qualitative examples in Figure 18, which includes (1) an object  casting a shadow across a floor-wall corner; and (2) an object placed on a curved surface. It shows that our proposed method correctly captures the distorted shadow shapes as results of these complex background geometries. These results further demonstrate the robustness of our proposed method.
> > >
> > > ---
> > >
> > > ## Ambiguous Cases (Fig. 3d)
> > > We thank the reviewer for this detailed geometric analysis. The binary case in Fig. 5(e) and ($\min | e^d | < \eta$) are included for illustrative purposes only. If the proposed LGI is binarised with a threshold, the ambiguous case in Figure 3(d) would indeed be classified as ``non-occlusion”. However, **the LGI is represented with continuous real values in practice**. As shown in Figure 4(b-d) where our proposed LGI embedding channels encode the occlusion range and the closest elevation distance in continuous real values. Because the model receives these continuous soft values rather than a discrete 0/1 mask, it is not limited by rigid criteria. Instead, it preserves the underlying geometric signal, allowing the network to learn and resolve these ambiguous boundary cases directly from the data.

---

### Official Review · Reviewer_RYL2 · 2025-10-30

**Soundness:** 2
**Presentation:** 2
**Contribution:** 2
**Rating:** 4
**Confidence:** 4

**Summary:**

This paper presents a novel shadow generation and relighting framework, which introduces LGI maps to provide lighting and shadow cue for the model. The proposed method derives the LGI maps from the estimated 2.5 D depth map and lighting conditions, then a LGI maps and lighting parameter conditioned bridge matching model generates the relit images .  Experiment results are provided to demonstrate the effectiveness of the framework, highlighting the good performance and efficiency of the proposed method.

**Strengths:**

1. This paper proposes a novel shadow generation and relighting framework. The LGI map can model the light-geometry interactions from the depth map which encodes useful shadow and illumination information, enabling the model to generate reasonable relighting results.
2. Extensive experiments show that the method achieve better performance than baselines.

**Weaknesses:**

1. I have some doubts about the proposed LGI map and whether it can accurately represent shadow and lighting information. Consider the point $p$ in fig.3 (a), and for this point, $e^d_n$ is always negative, which means that the elevation of light is always higher than the horizon, so this point can see the light source and its $c^m_1$ is negative. But for the point $p$ in fig.3 (c), this point is in the shadow, but its $c^m_1$ is also negative. So why the $c^m_1$ indicates the potential start of occlusion. And why is $c^m_1$ darker in the shadow area of fig.4(b)? Logically, in areas under a light source, $c^m_1$ should always be negative, meaning that the illuminated region should have a darker $c^m_1$.
2. The derivation of LGI maps and shadow mask does not consider the radius or the area of the light. When the light source is a single point, then we will obtain the hard shadow. But when the light source is not a single point, then we should consider soft shadow, including umbra and penumbra. It seems that the mixture of $c^m_1$, $c^m_2$ and $c^m_3$ can handle the soft shadow, but i am not very sure.
3. The cases in the dataset may be too simple, only a floor and a wall. Could the proposed method handle background with more complex geometry and texture?
4. The baseline used is not strong enough. Could the authors use more recent generative relighting methods, such as DiffusionRenderer and DiLightNet?

Liang, Ruofan, et al. "Diffusion Renderer: Neural Inverse and Forward Rendering with Video Diffusion Models." *Proceedings of the Computer Vision and Pattern Recognition Conference*. 2025.

Zeng, Chong, et al. "DiLightNet: Fine-grained lighting control for diffusion-based image generation." *ACM SIGGRAPH 2024 Conference Papers*. 2024.

**Questions:**

See weakness

---

> ### Author Response · Authors · 2025-11-25
>
> We sincerely thank the reviewer for the careful reading of our paper, for recognizing the novelty of our framework and the value of the proposed dataset, and for raising several insightful questions. Below, we address each concern in detail and provide additional visualization, and include the corresponding changes (highlighted in blue) in Sec.2 and App.J of the paper. We hope these clarifications resolve the main points of confusion and further support the contribution of our method.
>
> ---
>
> ## Interpretation of the LGI map
>
> For clarity, $e_n^d = e^s_n-e^l$ ​​is the angular deviation from the light ray. In Fig.3(a) $e^s_n$ and $e^l$ are all negative, because our normal vector is point aligned with the camera direction. For example, $e^s_1 = -\frac{1}{8}\pi$, $e^s_2 = -\frac{1}{16}\pi$, and $e^l = -\frac{1}{4}\pi$, as result $e_n^d = \( \frac{1}{8} \pi, \frac{3}{16}\pi \) $ all positive values, and $C^m_1 = min(e_n^d)$, choose the smallest value,  in fig .3(a) selects the red line passing through $S'_1$, which is why it indicates the start of occlusion. In Fig.3(c),  $e_1^d < 0$ and $e_2^d > 0$, so $C^m_1<0$, indicating that the point lies in shadow. Consistently, Fig. 4(b) visualizes negative $C^m_1$  (shadowed) as darker regions.
>
> ---
>
> ## Soft-shadow modeling
> Yes, LGI itself does not encode the light radius; the radius is introduced through the light parameters in $C^l$. LGI provides a distance-to-shadow boundary signal, which naturally helps model soft-shadow behavior.
>
> ---
>
> ## Simple Background
>
> Our method can be extended to more diverse settings. As shown in Fig.16 and Fig.17, where our method shows good generalization to complex backgrounds. Furthermore, unlike conventional object insertion methods, ours is the first capable of not only relighting foreground objects but also producing foreground-background lighting interactions such as shadows, inter-reflections, and refractions from two images (please refer to Fig.19). Handling two-way light transportation is an arguably more challenging problem setting for many downstream applications.
> For the harmonization setting,  our evaluations already cover datasets with substantial background complexity, such as non-planar geometry and rich textures in DESOBAv2 (see Fig. 17). Our strong performance on DESOBAv2 confirms that the method can robustly handle such complexity in practice.
> Despite the simpler backgrounds in ShadRel, joint shadow generation and object relighting remain challenging for objects with complex geometry and texture due to self-shadowing on intricate surfaces. Thus, our model is also optimized to handle such cases. As shown in Fig. 1 and Fig. 16, it produces high-quality results even with non-planar, textured, and cluttered backgrounds.
>
> ---
>
> ## Recent methods
>
> Our main baseline is a recent method published at ICCV 2025. DiffusionRenderer and DiLightNet are also strong relighting approaches, but they cannot be used as fair baselines due to data and setting mismatches.
>
> DiffusionRenderer parameterizes illumination using environment maps, whereas our setting focuses on explicit point-light control. This fundamental difference in illumination representation means it does not perform well in our setting without retraining. However, retraining is not feasible because DiffusionRenderer requires ground-truth normal, roughness, and metallic maps, which are unavailable in both datasets we use. In contrast, our method does not assume access to, or supervision from, ground-truth roughness, metallic, normal, or depth maps.
>
> DiLightNet likewise parameterizes illumination via environment maps and uses a two-stage pipeline (foreground object rendering followed by background inpainting) that is incompatible with our joint shadow-generation and relighting setup. More importantly, it targets a different problem: its input is a text prompt rather than an object–background pair.
>
> We have added a discussion of these methods in the related work section.

---

> > ### Comment · Reviewer_RYL2 · 2025-11-26
> > **Official Comment by Reviewer RYL2**
> >
> > I appreciate the efforts the authors have done, which make the paper more self-contained and easy to follow.
> >
> > The authors response addresses my questions and the additional experiment results demonstrate the effectiveness of the proposed method. To be specific,
> > 1. The authors clarification on the derivation of LGI map and the soft-shadow modeling confirms my belief that the proposed method can characterize complex shadows.
> > 2. The results in Fig.16 and Fig.17 demonstrate the effectiveness of the proposed method on background with complex texture and geometry for both real-world and synthetic cases.
> > 3. The authors also provide a detailed explanation of baseline selection.
> >
> > Based on the reasons mentioned above, I will raise my score to "accept" and thank the author for their efforts during the rebuttal.

---

> > > ### Author Response · Authors · 2025-11-26
> > >
> > > We are extremely grateful for your reconsideration and the decision to raise the score to 8.
> > >
> > > We are delighted to hear that our clarifications successfully addressed your concerns. We also appreciate your positive feedback on the additional results in Figures 16 and 17; we believe these experiments have significantly strengthened the validation of our method.
> > >
> > > Thank you for your constructive comments throughout this process, which have undoubtedly made the paper more self-contained and easier to follow.

---

### Official Review · Reviewer_FiRe · 2025-10-31

**Soundness:** 2
**Presentation:** 2
**Contribution:** 3
**Rating:** 6
**Confidence:** 3

**Summary:**

This work presents a new 2.5D representation from RGB images, depth information, and lighting parameters, termed as Light-Geometry Interaction maps (LGI), to boost the joint shadow generation and object relighting. As claimed in the paper, this LGI representation provides a physics-inspired prior to reduce floating and implausible shadow effects and empowers a neat pipeline to tackle direct illumination, secondary reflections, and inter-reflections. Besides, to enhance their pipeline training and evaluation, a synthetic dataset with 800K samples is constructed. Their experimental results illustrate better quantitative and qualitative performance compared to prior works.

**Strengths:**

1. LGI provides an intuitive utilization of the depth prior and suits the pipeline objective as joint shadow generation and object relighting.
2. The proposed pipeline integrates the modern large model backbones, such as depth estimation models and latent bridge matching models.
3. Dataset contribution: The provided dataset, ShadRel, provides a valuable contribution to this community.

**Weaknesses:**

1. Dependence on monocular depth information. A significant improvement of the proposed pipeline is introducing the depth information into the LGI representation, which also raises concerns. Monocular depth is an ill-posed estimation, and it frequently fails to provide accurate estimations, thus leading to malicious priors in the generation pipeline. This requires further clarification. At the same time, although LGI provides better performance compared to previous baselines, the depth information constitutes an extra input requirement not shared by several baselines (e.g., LBM, CSG), which only rely on RGB and light parameters.
2. The experimental results are weakened by the tight relationships between the proposed method and the ShadRel dataset. Although the authors have retrained baseline methods on the ShadRel dataset, this does not fully address the domain bias, since the ShadRel is synthetic and specifically tailored for the method’s assumptions and requirements (such as a single object, known light direction, and planar contact). This raises concerns that the illustrated improvements may primarily reflect the dataset alignment instead of model effectiveness.

**Questions:**

1. The potential impact of inaccurate depth estimation should be justified.
2. Enhanced ablation studies are recommended, which illustrate other ways to incorporate the depth information or inject this information into baseline methods. These studies are beneficial for validating the paper’s core design LGI. A fairer comparison would require facilitating baselines with similar depth guidance or reporting ablations isolating the effect of added depth input.
3. To validate model generality, it is recommended to evaluate models on real-image benchmarks using identical training data or provide cross-dataset results demonstrating robustness.

---

> ### Author Response · Authors · 2025-11-25
>
> We sincerely thank the reviewer for the detailed and constructive feedback, and for recognizing the novelty and dataset contribution of our work. Below, we clarify the main concerns, and we include the extended results (highlighted in blue)  in Sec.5.2 and App.H of the paper.
>
> ---
>
> ##  Potential Inaccuracy Monocular Depth and Comparison with Depth-Based Baselines
> Our LGI does not require metrically accurate depth; it only uses depth to derive coarse relative occlusion structure (e.g., near vs. far). As a result, global scale ambiguity and small local depth noise have limited impact on the LGI signal. In our ablation study, we compare our method using two different monocular depth estimators, and the performance is very similar across them. We further evaluate our method using ground-truth depth and observe no significant improvement over monocular depth. These results indicate that our pipeline is robust to typical monocular depth errors and that depth serves as a coarse geometric prior.
> ```
>            |            Overall          |     Shadow regio      |Object region
>            | RMSE    SSIM    BER   IOU   | RMSE    SSIM    BER   |RMSE    SSIM
> LBM+depth  |0.0395  0.7122 0.1003 0.7284 |0.1208  0.5923  0.1901 |0.0321  0.6692
> Ours       |0.0334  0.7227 0.0588 0.8096 |0.0898  0.6195  0.1103 |0.0282  0.6875
> Gt_depth   |0.0326  0.7266 0.0558 0.8107 |0.0894  0.6195  0.1070 |0.0280  0.6875
> ```
> First, in our ablation study (Table 4), we evaluate our method without using depth (line 462 of the updated paper) as input and with raw depth as input (line 464). The results show that raw depth is helpful, but still less effective than our LGI representation (line 463).
> Second, we additionally equip LBM with depth input (CSG does not provide public code, so a similar modification is not feasible). Our method consistently outperforms this depth-augmented LBM across all regions and metrics, demonstrating that the gains of LGI are due to the proposed design rather than simply the use of an extra depth modality.
>
> ---
>
> ## Real-image benchmark
>
> We agree that a real-image benchmark for light-controllable joint shadow generation and object relighting would further strengthen the paper. However, there is currently no public real-image dataset for this task, and constructing one is non-trivial and cannot be completed within the rebuttal period. Establishing a comprehensive real-image benchmark is therefore part of our planned future work.
>
> In addition, qualitative results on real images are included in the originally submitted supplementary video, and we further provide additional real-image visualizations, including multiple objects in the updated supplementary video. These qualitative results demonstrate that our method generalizes effectively to real-world images.
>
> Furthermore, the DESOBAv2 harmonization dataset consists of real images with multiple objects and non-planar backgrounds (see App. J of the updated paper). Our results on DESOBAv2 demonstrate that the proposed method performs well on real data.

---

> > ### Comment · Reviewer_FiRe · 2025-11-27
> >
> > Thanks for the efforts during the rebuttal period. The provided clarifications and additional results addressed my concerns, so I would like to raise my score to 8.

---

> > > ### Author Response · Authors · 2025-11-27
> > >
> > > We sincerely thank you for recognizing our efforts and for raising the score. We are glad that the provided clarifications and additional results successfully addressed your concerns. We appreciate the time you took to guide us in improving the paper.

---

### Official Review · Reviewer_YF9H · 2025-11-01

**Soundness:** 4
**Presentation:** 3
**Contribution:** 3
**Rating:** 8
**Confidence:** 4

**Summary:**

This paper proposes a unified pipeline that jointly generates shadows and performs object relighting, achieving physics-based control. It also introduces the ShadRel dataset for coupled light transport. The experimental setup is comprehensive, and the visual results are excellent for both synthetic and real-world cases.

**Strengths:**

- This paper presents the first generative method for the joint synthesis of shadows and object relighting from a single 2D RGB image. A key innovation is the introduction of a control mechanism conditioned on monocular depth, which enables the generation of continuous and coherent shadow and relighting effects in direct response to continuous variations in illumination.
- This paper introduces ShadRel, a large-scale synthetic dataset developed for the tasks of shadow generation and object relighting. The dataset is notable for its extensive volume and the inclusion of diverse and complex lighting scenarios, rendering it a valuable asset for training and evaluating models in these domains.
- This paper introduces a novel light-aware occlusion representation, LGI map. The methodology for acquiring this map is clearly articulated, and both theoretical analysis and experimental results compellingly demonstrate that the LGI map enables the generation of results with physically-based shadow and illumination constraints. This representation effectively encodes light-aware occlusion from monocular depth, providing a physics-inspired prior that constrains generative models to produce more realistic and consistent outputs.
- The paper is well-supported by thorough experimentation, clear theoretical explanations, and compelling visual results.

**Weaknesses:**

The visual results presented in the paper are compelling. However, the experiments primarily showcase objects with relatively simple geometric structures. To more rigorously assess the robustness and generalization capabilities of the proposed method, I would encourage the authors to include results on more structurally complex objects. For instance, objects with fine-grained details, intricate parts, or significant self-occlusion (e.g., a bicycle, a detailed sculpture, or a potted plant) would serve as more challenging test cases. Demonstrating high-fidelity shadow and relighting effects on such objects would provide stronger evidence of the model's effectiveness and further strengthen the paper's contributions.

**Questions:**

Please refer to the weakness.

---

> ### Author Response · Authors · 2025-11-25
>
> We sincerely thank the reviewer for the positive assessment and for the insightful suggestion regarding complex object geometries. We include the extended results in App.I of supplementary material.
> ## Complex Objects Results
> Our datasets already contain complex objects, such as bicycle, potted plant. We construct a 200-sample subset consisting only of these challenging cases and evaluate our method on this subset. On this subset, our method outperforms LBM by a large margin, indicating that our approach is robust beyond simple shapes.
> ```
>      |            Overall           |     Shadow regio      |Object region
>      | RMSE    SSIM    BER   IOU    | RMSE    SSIM    BER   |RMSE    SSIM
> LBM  |0.0431 0.7112  0.0954  0.0691 |0.1575  0.5609  0.1758 |0.0342  0.6743
> Ours |0.0341 0.7196  0.0658  0.7940 |0.0929  0.6114  0.1241 |0.0286  0.6828
> ```
> We also provide qualitative visualizations of these complex objects in both synthetic and real images, which are included in the updated supplementary video.
> Moreover, the DESOBAv2 dataset also includes complex objects, such as humans with bicycles; our results on this dataset (Fig.7) likewise show that the method preserves high-quality predictions in these cases, further supporting its robustness to complex geometries.

---

### Meta-Review · Area_Chair_9N2p · 2026-01-11

**Summary:**

This paper introduces Light–Geometry Interaction (LGI) maps, a physically-based representation derived from monocular depth that enables joint shadow generation and object relighting in a generative framework. Reviewers agree that the representation is well-motivated and addresses a meaningful gap. Initial concerns regarding dependence on monocular depth, dataset bias, generalization to complex geometry and the clarity of the LGI formulation were largely addressed through the author response. Remaining concerns focus primarily on the degree of conceptual novelty relative to prior graphics-based approaches and the limited exploration of certain experimental settings. Given that the major technical concerns are sufficiently addressed, that the method is well-considered and leads to an empirically strong representation for shadow generation and relighting, the paper is recommended for acceptance to ICLR. It is suggested that the authors incorporate all reviewer suggestions in the final version of the paper.

**Reviewer Concerns:**

### Addressed concerns
* **YF9H:** The evaluation focuses mainly on objects with simple geometry and may not demonstrate robustness to complex structures. The author response adds experiments on complex objects (e.g., bicycles, potted plants, and human-object cases) and reports consistent quantitative and qualitative improvements, addressing the reviewer’s concern.
* **FiRe:** The method may rely too heavily on monocular depth and performance gains might be attributable to the added modality rather than the LGI design. The author response provides ablations with multiple depth estimators, ground-truth depth and depth-augmented baselines, showing that LGI’s formulation rather than raw depth input accounts for the improvements.
* **FiRe:** Results may be biased by tight coupling to the synthetic dataset used for training and evaluation. The author response adds cross-dataset experiments and real-image evaluations, demonstrating that gains persist outside the synthetic distribution.
* **RYL2:** The correctness of LGI representation, its ability to model soft shadows and handle complex backgrounds are initially unclear. The author response provides a clearer mathematical explanation of LGI construction and experiments on textured and non-planar backgrounds, resolving the technical questions raised.

### Unaddressed concerns
* **2xqw:** The conceptual novelty of LGI is limited, as the representation is closely related to existing ray-based or graphics-inspired occlusion modeling approaches. The author response emphasizes incorporation into a generative framework but does not directly establish the core idea as significantly new.
* **2xqw:** The experimental setting is somewhat constrained, with the primary focus on point-light sources and simplified illumination scenarios. While the author response points to supplementary results and secondary tasks, it does not fully demonstrate broad coverage.

**Reviewer Scores:**

* **YF9H:** Initial rating 8, all concerns addressed, would likely maintain 8.
* **FiRe:** Initial rating 8, added results would likely increase to 8.
* **RYL2:** Initial rating 8, added experiments and conceptual explanations would likely raise to 6 or 8.
* **2xqw:** Initial rating 2, some clarifications on novelty and experiments would likely raise to 4.

---

### Decision · Program_Chairs · 2026-01-26

Accept (Poster)